# Immune synapse formation promotes lipid peroxidation and MHC-I upregulation in licensed dendritic cells for efficient priming of CD8+ T cells

Diego Calzada-Fraile [1], Salvador Iborra [2], Marta Ramírez-Huesca[1], Inmaculada Jorge[1,3], Enrico Dotta[4], Elena Hernández-García[2], Noa Martín-Cófreces [3,5], Estanislao Nistal-Villán [6], Esteban Veiga [7], Jesús Vázquez [1,3], Giulia Pasqual [4,8] & Francisco Sánchez-Madrid [1,3,9] ✉

Antigen cognate dendritic cell (DC)-T cell synaptic interactions drive activation of T cells and instruct DCs. Upon receiving CD4+ T cell help, post-synaptic DCs (psDCs) are licensed to generate CD8+ T cell responses. However, the cellular and molecular mechanisms that enable psDCs licensing remain unclear. Here, we describe that antigen presentation induces an upregulation of MHC-I protein molecules and increased lipid peroxidation on psDCs in vitro and in vivo. We also show that these events mediate DC licensing. In addition, psDC adoptive transfer enhances pathogen-specific CD8+ T responses and protects mice from infection in a CD8+ T cell-dependent manner. Conversely, depletion of psDCs in vivo abrogates antigen-specific CD8+ T cell responses during immunization. Together, our data show that psDCs enable CD8+ T cell responses in vivo during vaccination and reveal crucial molecular events underlying psDC licensing.

Cross-talk between T cells and DCs is one of the foundation pillars of adaptive immunity. This interaction serves to prime CD4+ T cells, but it also licenses DCs for the generation of CD8+ T cell responses[1,2]. Licensing is crucial for tumor rejection[3] and resolution of infectious processes[4]. CD4+ T cell help is provided through antigen presentation to DCs via CD40 signaling[5–7]. DC licensing lowers the challenge threshold to elicit primary CD8+ T cell responses[4,8,9] although in some cases strong innate pathogen-induced signaling can bypass it[10]. CD40 and TLR signaling act as a synergistic pair that underlies DC licensing[11]. However, CD4+ T cell help is needed for secondary responses and the maintenance of memory CD8+ T cells[12–15]. DC licensing is required for antigens to be cross-presented to CD8+ T cells[1,6,7], which is particularly important for antitumor and some anti-pathogen CD8+ T cell responses[16].

The structure formed during cognate interactions between T cells and DCs, commonly known as the immune synapse, serves as a platform for the transfer of information from DCs towards T cells, leading to their activation[17]. However, recent studies have indicated that communication is not purely unidirectional. There is now evidence

[1]Centro Nacional de Investigaciones Cardiovasculares, 28029 Madrid, Spain. [2]Department of Immunology, Ophthalmology and ENT, School of Medicine, Universidad Complutense de Madrid, 28040 Madrid, Spain. [3]Centro de Investigación Biomédica en Red de Enfermedades Cardiovasculares (CIBERCV), 28029 Madrid, Spain. [4]Laboratory of Synthetic Immunology, Department of Surgery, Oncology and Gastroenterology, University of Padova, Padova, Italy. [5]Dynamic Video Microscopy Unit, Immunology Department, Instituto de Investigación Sanitaria Hospital Universitario La Princesa, Universidad Autónoma de Madrid, 28006 Madrid, Spain. [6]Microbiology Section, Departamento CC, Farmacéuticas y de la Salud, Facultad de Farmacia, Universidad CEU San Pablo, Boadilla del Monte, 28668 Madrid, Spain. [7]Department of Molecular & Cellular Biology, Centro Nacional de Biotecnología (CNB-CSIC), Madrid, Spain. [8]Veneto Institute of Oncology IOV-IRCCS, Padua, Italy. [9]Immunology Department, Instituto de Investigación Sanitaria Hospital Universitario La Princesa, Universidad Autónoma de Madrid, 28006 Madrid, Spain. ✉e-mail: fsmadrid@salud.madrid.org

that immune synapses trigger signals also on the DC-side. Specific information travels from the T cell to the antigen-presenting cell (APC) through membrane-receptor interactions or within extracellular vesicles, influencing the functional outcome of the APC[18,19]. Novel approaches have allowed to describe that immune synapse formation instructs post-synaptic DCs (psDCs) to undergo a transcriptomic reprogramming[20–23] and epigenomic remodeling that increases their migratory ability[20]. Also, immune synapses increase DC survival[24], which may explain the "memory-like" phenotypes observed in DCs[25]. However, increased survival alone does not explain their greater ability to induce CD8[+] T cell responses[26]. Others and we postulate that T cells may functionally reprogram DCs through the immune synapse formation, including licensing. However, the mechanisms governing this process remain unknown.

In this work, we address the specific mechanisms that enable DC licensing for efficient antigen cross-presentation to CD8[+] T cells upon cognate contacts with CD4[+] T cells.

## Results

### Antigen presentation induces psDCs proteomic remodeling to increase MHC class I molecules

To address the molecular modifications induced by immune synapse formation within DCs, we performed an unbiased mass spectrometry assay in DCs using an in vitro antigen presentation model to generate postsynaptic DCs (psDCs) or nonsynaptic DCs (nsDCs) (Fig. 1a). First,

we experimentally validated immune synapse formation by confocal microscopy in psDCs. T cells interacting with DCs displayed F-actin lamellas and centrosome polarization towards the contact with psDCs but not with nsDCs (Supplementary Fig. 1a). The proteins of psDCs and nsDCs were subjected to high-throughput quantitative proteomics using multiplexed isobaric labeling followed by LC-MS/MS. A total of 3508 proteins were detected and quantified (FDR < 0.01) (Supplementary Data 1). Partial least squares discriminant analysis (PLS-DA) indicated a good separation between nsDC and psDC samples, albeit the samples from nsDC group displayed higher inter-sample heterogeneity (Supplementary Fig. 1b). Analysis of protein abundance changes revealed 23 proteins differentially expressed in psDCs vs nsDCs (FDR ≤ 0.05; number of peptides ≥ 2), of which 6 (26%) were more abundant and 17 (74%) were less abundant in psDCs (Supplementary Data 1). Next, we performed an unbiased analysis of abundance changes in functional categories produced by coordinated protein alterations[27] (Supplementary Data 2). Data were represented using heatmap and sigmoid graphs of the changes in relative abundance of proteins comprising specific categories to identify groups of proteins that were up- or downregulated in a coordinated manner. Such analysis revealed that the eukaryotic ribosomal proteins were downregulated while the mitochondrial ones remained unchanged (Fig. 1b and Supplementary Fig. 1c). Histones were also downregulated compared to histone-interacting proteins (Supplementary Fig. 1d). Also, proteins related to glucose and lipid metabolism and the maintenance

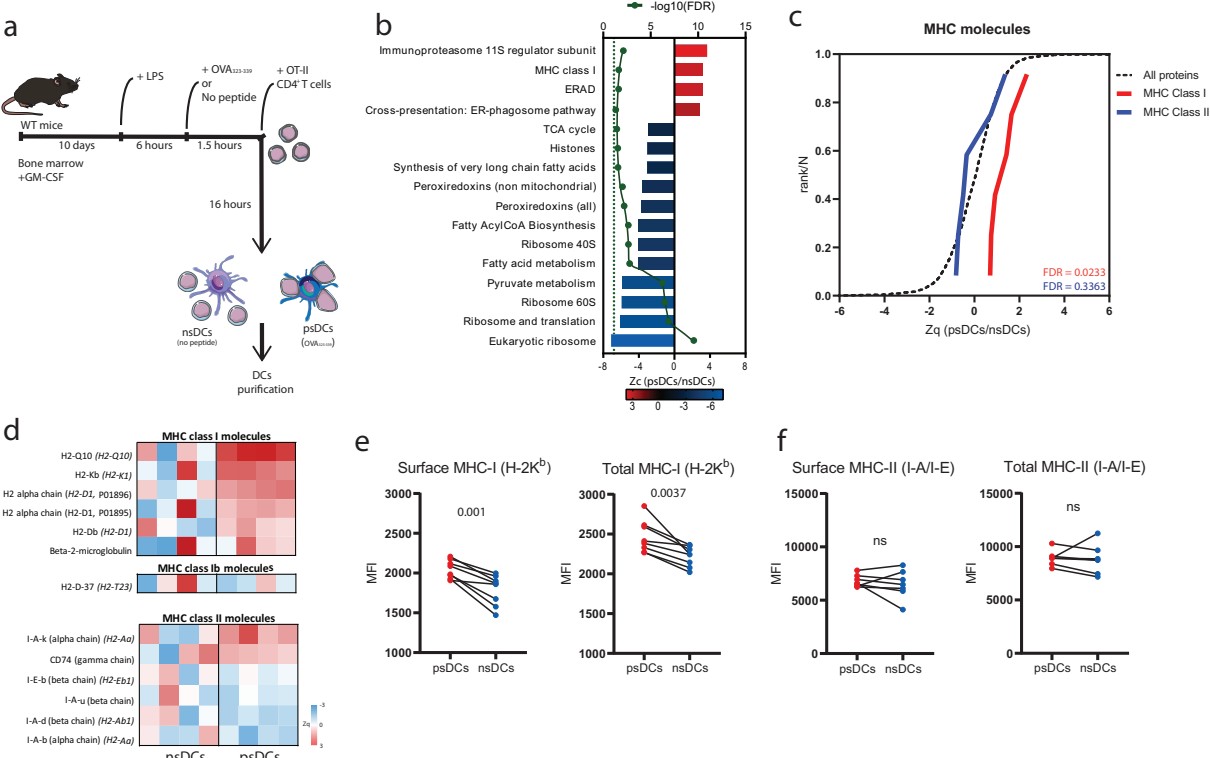

**Fig. 1 | Protein pattern reprogramming of psDCs. a** Schematic representation of the protocol for the generation of psDCs and nsDCs. **b** Selection of protein functional categories found to be up- or downregulated in psDCs compared to nsDCs (*n* = 4 biological replicates). Bars in blue/red color show category abundance change Zc, which is the log₂ fold change in psDCs compared to nsDCs samples expressed in units of standard deviation. Green dots represent the -log₁₀(FDR) value of the statistically significant abundance changes. Dashed green line is the cutoff for an FDR < 0.05. **c** Coordinated behavior of the identified proteins belonging to MHC class I (red line) and MHC class II categories (blue line) compared to all proteins (dashed black line). Sigmoid plots show the distribution of the protein abundance changes Zq, which is the log₂ fold change in psDCs vs. nsDC,

expressed in units of standard deviation. FDR statistic values of the category abundance changes are indicated. **d** Heatmap representation of the protein abundance changes (Zq) in all the nsDC and psDC replicates respect to the average value of nsDC samples, classified into MHC class I, class Ib and class II categories. **e** Flow cytometry analysis of surface (left) or surface and intracellular levels (right) of MHC class I (H-2K^b) molecules obtained by flow cytometry in psDCs and nsDCs after coculture, *n* = 8 biological replicates. **f** Same analysis as (**e**) for MHC class II (I-A/I-E) molecules in their surface (left) or surface and intracellular levels (right), *n* = 7 biological replicates. For (**e**, **f**) data is representative of two independent experiments. Statistical significance tests used are two-tailed paired *t*-tests **e**, **f**. *p*-values are indicated, with ns for *p*-value > 0.05.

of redox homeostasis were downregulated (Fig. 1b). Conversely, categories linked to antigen presentation to CD8 + T cells were upregulated, including MHC class I molecules, the ER-phagosome pathway of cross-presentation, the immunoproteasome 11 S regulator subunit and endoplasmic-reticulum-associated protein degradation (ERAD) proteins (Fig. 1b). In fact, most of the ERAD pathway displayed a coordinated upregulation (Supplementary Fig. 1e). This pathway is involved in protein release from the ER lumen to the cytosol, thus it is important in antigen cross-presentation[28]. Interestingly, most proteasome subunits remained unchanged except for the 11 S subunit which belongs to the immunoproteasome (Supplementary Fig. 1f), which is involved in antigen processing and presentation via the MHC class I route[29]. In agreement with this, MHC class I molecules were significantly upregulated, while MHC class II molecules remained unchanged (Fig. 1c, d). Interestingly, H2-D-37, an MHC class Ib molecule (H2-Qa-1) that is usually present in low amounts[30], remained unchanged. To independently corroborate these findings, we stained psDCs with specific antibodies and performed flow cytometry experiments. H-2K[b], an MHC class I molecule of the C57BL/6 mice haplotype, was both increased on the surface and total psDCs compared to nsDCs (Fig. 1e). Conversely, the surface and total levels of I-A/I-E (MHC class II molecules) remained unchanged (Fig. 1f). Globally, the proteomic remodeling induced by antigen presentation on DCs reveals specific protein groups that underlie the possible mechanisms through which DCs are licensed upon immune synapse formation with CD4[+] T cells, including molecular mediators of antigen cross-presentation and MHC class I molecules.

## Formation of immune synapses with CD4 + T cells enhances the ability of psDCs to induce antigen cross-presentation to CD8[+] T cells

CD4[+] T cell help is required for "classical" DC licensing for priming CD8[+] T cells[5–7]. To assess the differential abilities of psDCs to activate CD8[+] T cells, we cocultured purified psDCs or nsDCs that had been fed soluble OVA protein with effector OT-I CD8[+] T cells generated in vitro (Fig. 2a). Upon addition of OVA, psDCs displayed greater ability to cross-present OVA peptides to effector OT-I CD8[+] T cells (Fig. 2b). In

vivo, DCs may also encounter naïve CD8[+] T cells. To assess whether psDCs boosted activation of naïve CD8[+] T cells, we cocultured OVA-fed psDCs or nsDCs with resting OT-I CD8[+] T cells. As observed for effector cells, psDCs had an enhanced ability to activate resting CD8[+] T cells, boosting their proliferation (Fig. 2c, d), and production of IL-2 (Fig. 2e). Hence, DCs enhance activation of both effector and naïve CD8[+] T cell via cross-presentation following productive immune synapse formation with CD4[+] T cells.

## Lipid peroxidation mediates licensing for cross-presentation in psDCs

The molecular mechanisms that allow endocytosed antigens to access the cytosol for degradation and localization to MHC class I-loading sites remain incompletely understood[31]. Some of these mechanisms include lipid modifications. Ingenuity Pathway Analysis (IPA) of the proteomics changes observed in psDCs (Supplementary Data 3) predicted an upregulation of the ontology for accumulation of lipids in psDCs (Fig. 3a). Indeed, by using a probe to detect lipid droplets, which have a key role in cross-presentation[32–34], psDCs showed increased levels of fluorescence (Fig. 3b). Furthermore, we found that psDCs have increased levels of CD36, a scavenger receptor that mediates fatty acid uptake (Fig. 3c). These data agreed with the observed downregulation of several glucose and lipid metabolism routes (Fig. 1b), particularly pyruvate metabolism and fatty acid biosynthesis, which displayed a prominent coordinated behavior of downregulation (Supplementary Fig. 2a, b). Seahorse XF analysis revealed a decreased activity of the mitochondria, but no differences in glycolytic activity of psDCs (Supplementary Fig. 2c, d), as the proteomic dataset indicated (Supplementary Fig. 2a).

Disruption of the oxidative stress balance results in lipid peroxidation of endosomes. This alters the structure and integrity of the endosomal membrane, allowing antigen leakage to the cytosol for cross-presentation[35,36]. Functional category analysis of our proteomic data indicated a potential dysregulation of the redox hemostasis due to a decreased level of peroxiredoxins (Fig. 1b). Indeed, some of these proteins (non-mitochondrial peroxiredoxins) were coordinately downregulated. Interestingly, thioredoxins, which recycle oxidized

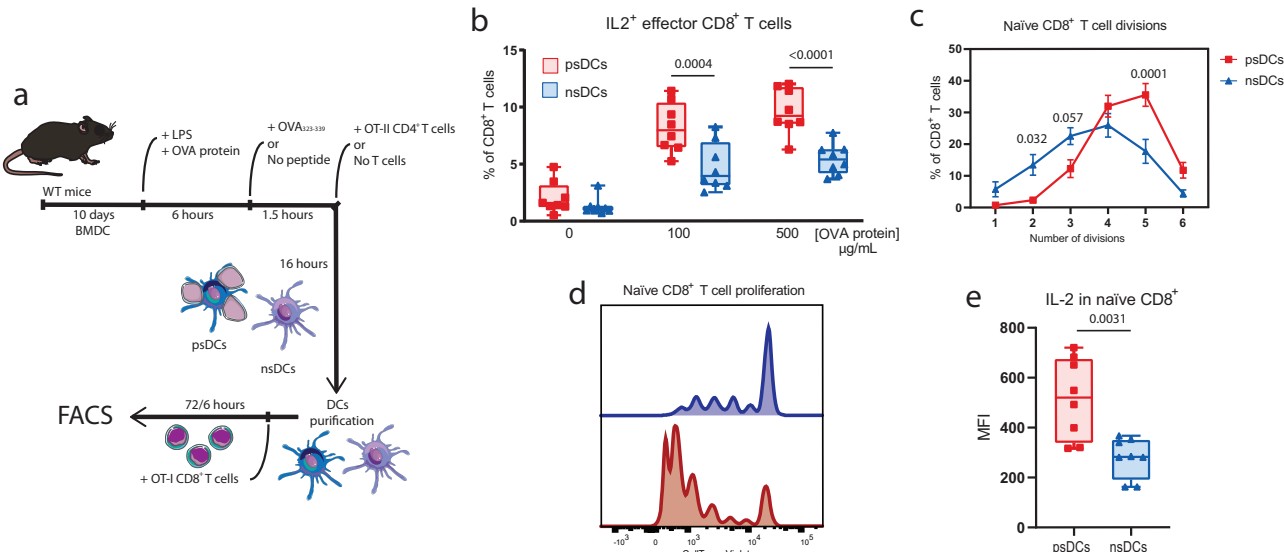

**Fig. 2 | Increased efficiency of cross-presentation to CD8[+] T cells. a** Experimental setup for cross-presentation assays. In all cases (**b–e**) $n = 8$ biological replicates. **b** Proportions of IL-2[+] cells among the effector OT-I CD8[+] T cells after 6 h coculture with psDCs (red) or nsDCs (blue) that had been fed with the indicated concentration of OVA protein. **c** Proportion of naïve OT-I CD8[+] T cells that have undergone the indicated number of divisions after 72 h coculture with psDCs or nsDCs fed with

500 μg/mL of OVA protein in the total OT-I CD8[+] T cell population, indicating mean ± SEM. **d** Proliferation of naïve OT-I CD8[+] T cells activated by psDCs (red) or nsDCs (blue). **e** Intracellular staining for IL-2 in naïve OT-I CD8[+] T cells after 72 h of coculture with psDCs or nsDCs. For b-e data is representative of two independent experiments. Statistical significance tests used are two-tailed paired t-tests (**e**) or two-way ANOVA (**b, c**). Error bars in (**c**) indicate mean ± SEM. p-values are indicated.

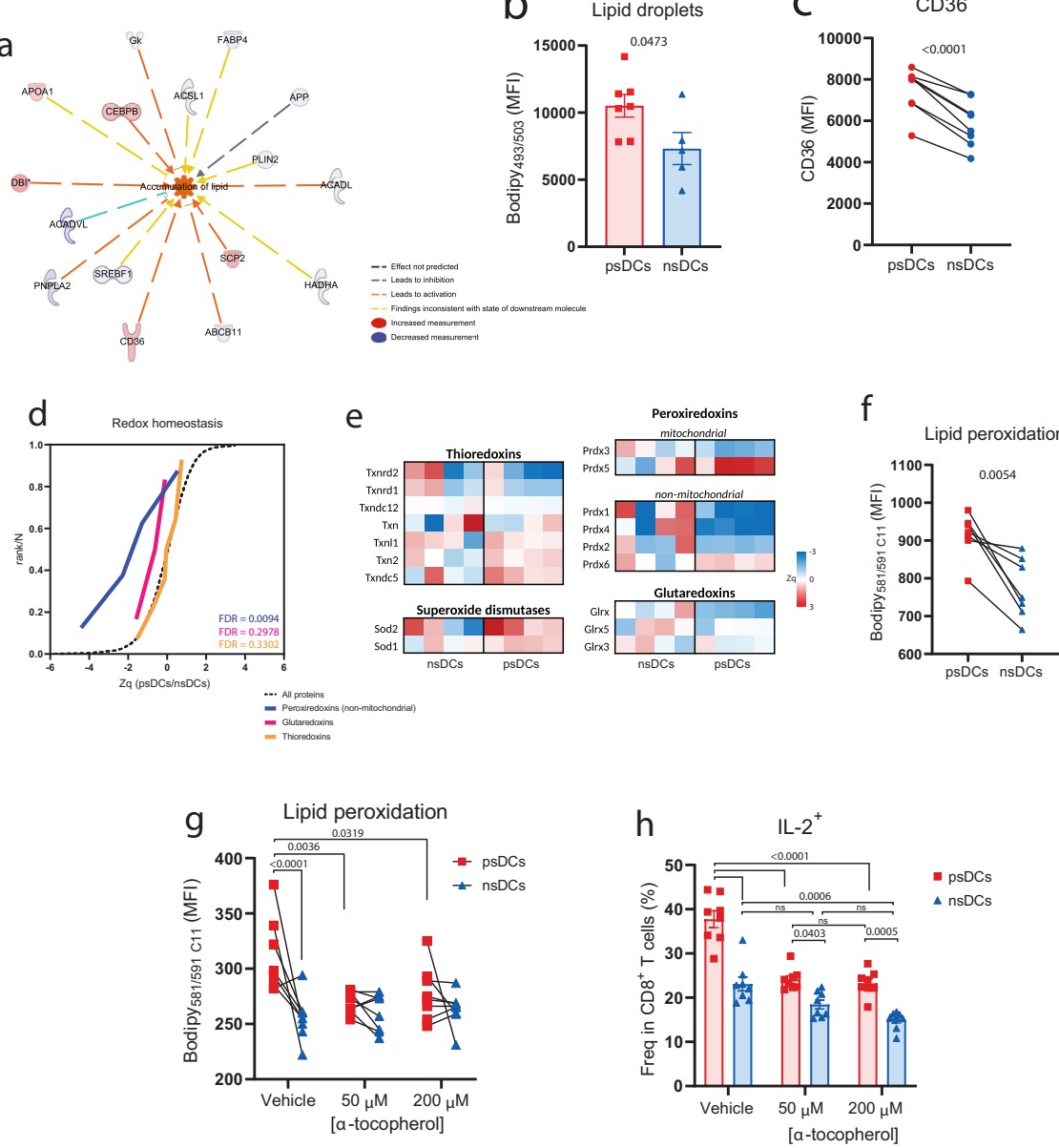

**Fig. 3 | Proteomic dysregulation of redox homeostasis increases lipid peroxidation for efficient cross-presentation to CD8⁺ T cells. a** IPA protein network of predicted upregulation of the category *accumulation of lipid*. **b** FACS analysis neutral and non-polar lipids using the BODIPY 493/503 probe in psDCs (*n = 7*) and nsDCs (*n = 5* biological replicates). **c** FACS analysis of the surface molecule CD36 in psDCs and nsDCs (*n = 6* biological replicates). **d** Quantitative proteomics analysis of proteins related to redox homeostasis categories. The distributions of quantitative protein values (Zq) in the comparison of psDCs vs. nsDCs are plotted. FDR statistic values of the category abundance changes are indicated. **e** Heatmap representation of redox homeostasis proteins abundance changes (Zq) in all the nsDC and psDC replicates respect to the average value of the nsDC samples. **f** FACS analysis of lipid peroxidation using the BODIPY 581/591 C11 lipid peroxidation probe. Data show the mean fluorescence intensity (MFI) in the B530/530 channel (*n = 7* biological replicates). **g** FACS analysis of lipid peroxidation in psDCs and nsDCs generated in the presence of α-tocopherol or vehicle (*n = 8* biological replicates). **h** Proportions of IL-2+ cells among effector OT-I CD8 + T cells obtained by FACS. Cells were analyzed after 6 h of coculture with psDCs or nsDCs generated in the presence of the indicated concentrations of α-tocopherol or vehicle (*n = 8* biological replicates). All data are representative of at least two independent experiments. Bar plots indicate mean ± SEM in all cases. Statistical significance was obtained with two-tailed unpaired (**b**) or paired *t*-tests (**c, f**) or two-way ANOVA (**g–h**). MFI: mean fluorescence intensity. *p*-values are indicated, with ns for *p*-value > 0.05.

peroxiredoxins remained unchanged whilst glutaredoxins displayed a coordinated downregulation (Fig. 3d, e). These data were consistent with a reduced capacity of psDCs for preventing peroxidation. In fact, we observed a significantly increase in oxidized peptides in psDCs by proteomics. Hence, we ascertained whether redox dysregulation was affecting lipids as well. By using a lipid peroxidation probe, we observed that psDCs displayed increased levels of lipid peroxidation compared to nsDCs, indicating that CD4⁺ T cell help promotes lipid peroxidation on DCs (Fig. 3f). As lipid peroxidation allows endosomal

antigen to escape for cross-presentation[36], we tested whether inhibition of lipid peroxidation abrogated the increased ability of psDCs to cross-present to CD8⁺ T cells. Therefore, we fed soluble OVA to psDCs or nsDCs and cocultured them with OT-II CD4⁺ T cells in the presence of different concentrations of α-tocopherol (vitamin E), a potent antioxidant. This resulted in the elimination of the lipid peroxidation differences between psDCs and nsDCs (Fig. 3g). Interestingly, α-tocopherol-treated psDCs had reduced abilities compared to non-treated psDC and nsDCs in terms of CD8⁺ T cell activation (Fig. 3h).

Therefore, a dysregulation of the redox homeostasis on psDCs resulted in increased levels of lipid peroxidation, which associated with the increased ability of psDCs to cross-present soluble protein to CD8[+] T cells.

## Immune synapsis induces MHC class I and lipid peroxidation in DC ex vivo and in vivo during immunization

BMDCs are a useful tool to study licensing of antigen presenting cells[2]. They share with DCs isolated from tissues the ability to present antigens to T cells and to respond to microbial stimuli through maturation, but they are considered as surrogates of in vivo-occurring conventional DCs (cDCs)[37,38]. Therefore, to validate our findings on primary DCs, we used the LIPSTIC (Labeling Immune Partnerships by SorTagging Intercellular Contacts) model, a proximity-based intercellular labeling method. In this model, CD40L is fused to the *S. aureus* transpeptidase A (SrtA) on CD4[+] T cells, which covalently transfers a biotinylated peptide to a polyglycine chain fused to the CD40 surface molecule of DCs upon cognate interaction of both cells. Therefore, DCs that interact with T cells on an antigen-dependent manner, psDCs, can be identified as they become positive for biotin[23]. Using an ex vivo antigen presentation assay, splenic DCs from $Cd40^{GS}$ mice were activated with LPS and pulsed with OVA$_{323-339}$ or LCMV GP$_{61-80}$, cocultured with OT-II CD4[+] T cells from CD4-Cre $Cd40lg^{Srta}$ mice (Fig. 4a) and analyzed by flow cytometry (Supplementary Fig. 3a). As expected, biotin[+] cells were observed only when DCs were pulsed with OVA$_{323-339}$ both in the DC (Fig. 4b) and CD4[+] T cell fraction (Supplementary Fig. 3b). Then, we analyzed the levels of H-2K$^b$ on the DCs. When subdividing the populations according to their biotin staining, we observed that H-2K$^b$ levels were significantly upregulated in the biotin[+] population of the OVA$_{323-339}$-pulsed DCs compared to biotin[-] DCs from the OVA$_{323-339}$- and LCMV GP$_{61-80}$-pulsed sample, which had similar levels of H-2K$^b$ to biotin[-] DCs from the OVA$_{323-339}$-pulsed sample (Fig. 4c). Likewise, lipid peroxidation was specifically increased in the biotin[+] population of the OVA$_{323-339}$-pulsed DCs (Fig. 4d).

Next, we assessed whether the same molecular changes in vivo by using the LIPSTIC mice to track in vivo postsynaptic or non-synaptic migratory DCs (mDCs) in the context of footpad immunization. After transfer of OT-II CD4[+] T cells from CD4-Cre $Cd40lg^{Srta}$ mice, $Cd40^{GS}$ mice were immunized with OVA:Alum or PBS:Alum and mDCs coming from the draining popliteal lymph nodes (pLNs) were retrieved and analyzed by FACS 72 h after immunization (Fig. 4e and Supplementary Fig. 3c). As expected, biotin staining in both the DC and CD4[+] T cell fraction only occurred when immunizing with OVA:Alum and not with PBS:Alum (Fig. 4f and Supplementary Fig. 3d), indicating that biotin staining does require antigen presentation. Immunization did not alter the proportions of mDCs (Supplementary Fig. 3e) or resident DCs (rDCs) (Supplementary Fig. 3f) and, as previously reported[23], biotin staining was predominantly observed in the mDC but not rDC population (Supplementary Fig. 3g). Interestingly, psDCs (biotin[+] mDCs) had significantly higher levels of H-2K$^b$ compared to both nsDCs (biotin[-]) from the same pLNs and from pLNs from mice injected with PBS:Alum. In contrast, no differences were observed when comparing nsDCs (biotin[-]) from mice immunized with PBS:Alum or OVA:Alum (Fig. 4g). We next analyzed lipid peroxidation in sorted biotin[+] and biotin[-] mDCs from OVA:Alum or PBS:Alum-immunized mice (Supplementary Fig. 3h-i). Likewise, lipid peroxidation increased in biotin[+] mDCs from the OVA-immunized pLNs (psDCs) compared to biotin[-] mDCs (nsDCs) from the same pLNs or mDCs coming from pLNs of the PBS:Alum group, which did not show any difference between them (Fig. 4h).

As CD40 was previously reported to be central to DC licensing[7], we next assessed the contribution of CD40 signaling to MHC-I and lipid peroxidation in psDCs. To do so, we blocked or not CD40L during an ex vivo antigen presentation assay with primary splenic DCs (Supplementary Fig. 3j). The differences of MHC-I levels between psDCs and nsDCs were not abrogated although the amount of MHC-I molecules significantly decreased in both psDCs and nsDCs when CD40-CD40L interactions were blocked during synaptic interactions (Supplementary Fig. 3k). Remarkably, lipid peroxidation significantly decreased in psDCs when blocking CD40 induction by CD40L, also displaying lower lipid peroxidation than nsDCs (Supplementary Fig. 3l). Surprisingly, nsDCs generated in the presence of anti-CD40L had increased levels of lipid peroxidation compared to the rest of conditions.

Hence, increased levels of MHC-I and lipid peroxidation occur in psDCs in a model of productive immune synapse formation ex vivo and during in vivo DC:T cell interactions upon footpad immunization. Moreover, CD40 induction contributes to MHC-I upregulation and increased lipid peroxidation induced by productive immune contacts in psDCs.

## psDCs protect mice from Listeria monocytogenes infection in a CD8[+] T cell-dependent manner

IPA analysis of the proteomics signature of psDCs predicted a prominent role during pathogen infection (Fig. 5a). Since psDCs displayed an enhanced ability to activate CD8[+] T cells in vitro, we wondered whether psDCs were also able to enhance CD8[+] T cell responses in vivo. To test this, we injected psDCs or nsDCs into WT mice and then infected them with *Listeria monocytogenes*. Remarkably, mice transferred with psDCs displayed significant increased survival upon administration of a lethal dose of *Listeria monocytogenes* than those transferred with the same number of nsDCs. Interestingly, the mere transfer of LPS-activated DCs showed a protective effect compared to the PBS group (Fig. 5b). Hence, we wondered whether the differences in survival were due to an early innate control of bacterial infection provided by psDCs. We assessed bacterial load in liver and spleen at early timepoints after infection. We observed no significant differences between mice that had been transferred psDCs or nsDCs, albeit an increase in bacterial load in mice injected with PBS at 3 days post-infection. This indicated that increased survival provided by psDCs compared to nsDCs was not caused by early control of the bacterial burden (Fig. 5c). The different groups did not display differences in weight loss before mice started to succumb to infection (Fig. 5d).

Next, we assessed whether protection from infection depended on adaptive mechanisms. We challenged $Rag1^{-/-}$ mice transferred with psDCs, nsDCs or PBS with a lethal dose of *L. monocytogenes*. Interestingly, the PBS group succumbed quicker to infection, corroborating the innate-mediated effect of LPS-activated DC transfer. However, we observed no significant differences in survival dynamics between $Rag1^{-/-}$ mice injected with psDCs or nsDCs (Supplementary Fig. 4a, b), indicating that an adaptive mechanism accounted for the increased survival provided by psDCs in wild-type mice. Activated CD4[+] T cells from the psDCs fraction secrete a range of cytokines that can modulate the immune response, which is relevant in *Listeria* infection[39]. Hence, to rule out that the protective effect of psDCs was due to the presence of OT-II CD4[+] T cells that are not removed during purification, we administered a number of purified CD4[+] T cells from the psDC fraction equal to of the number present in the psDC fractions and compared its effect to psDCs and PBS administration. Mice transferred with psDCs displayed increased survival compared to mice injected with only activated CD4[+] T cells, which survived to a comparable rate to that of mice injected with PBS (Supplementary Fig. 4c, d). This indicated that the protective effect of psDCs cannot be attributed to the presence of residual activated CD4[+] T cells in the psDC fraction, and that psDCs themselves accounted for it. We next tested whether protection from fatal infection relied on the increased abilities of psDCs to activate CD8[+] T cells, which have been reported to be critical for *L. monocytogenes* primary infection and generation of protective immunity[40,41]. psDCs, nsDCs or PBS were transferred to mice depleted of CD8[+] T cells using a monoclonal antibody, and infected with a lethal dose of *L.*

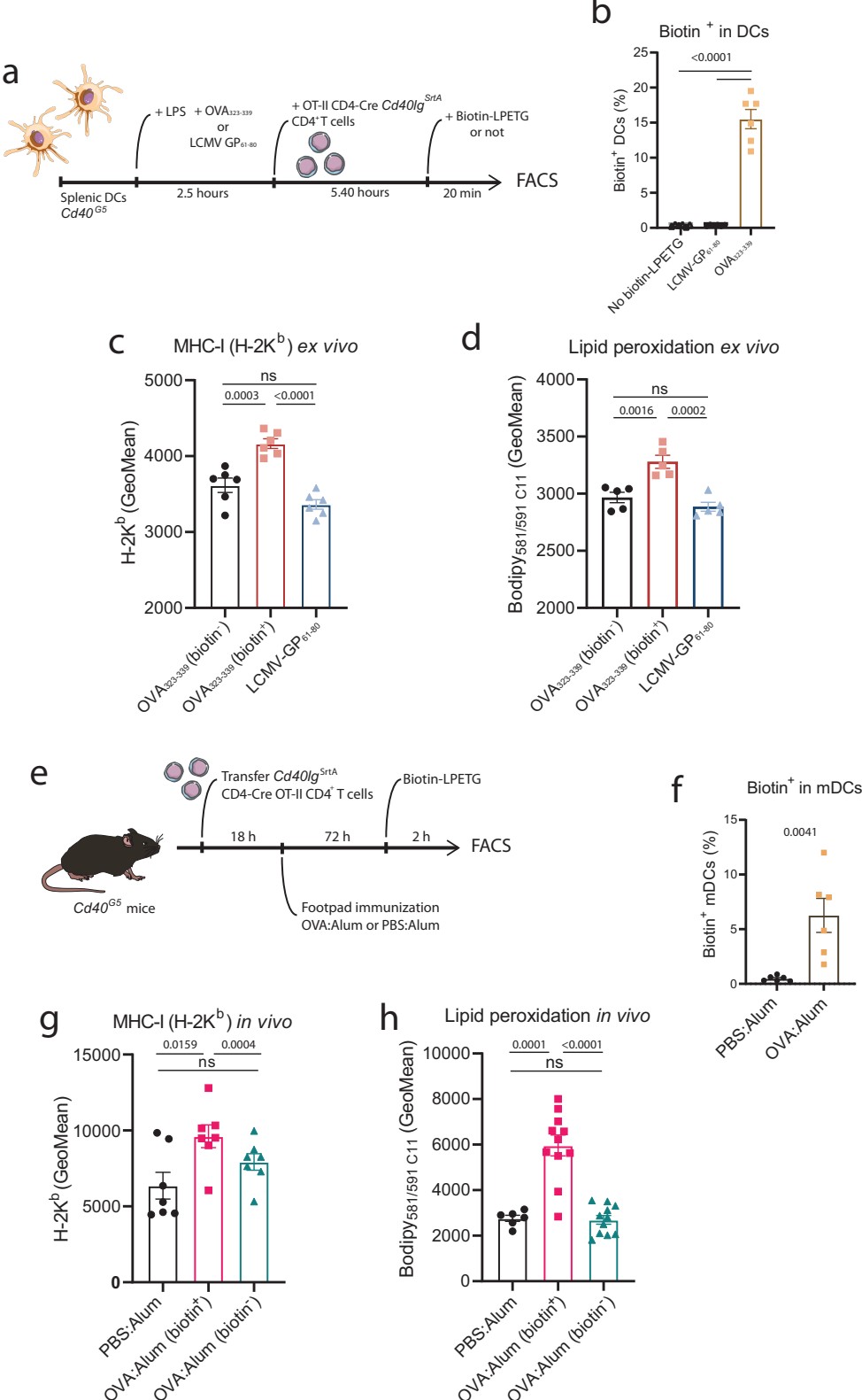

*monocytogenes*. We monitored that CD8+ T cells were absent during the whole course of infection (Supplementary Fig. 4e). As postulated, CD8+ T cell depletion abrogated the protective effect endowed by psDC transfer compared to nsDCs (Fig. 5e, f). To ascertain the location where the interaction between psDCs and CD8 + T cells could be taking place, we assessed the biodistribution of CD45.1+ psDCs, nsDCs or PBS injected in CD45.2+ mice. Both psDCs and nsDCs injected i.v. were

retrieved in the lung 24 h after injection, whereas we could not detect them in lymphoid tissues such as the spleen or lung-draining mediastinal lymph nodes (Supplementary Fig. 4f). Hence, to rule out the possibility that psDCs could be indirectly inducing CD8+ T cell activation by endogenous DCs in there or other tissues, we used $\beta_2$-microglobulin knockout psDCs or nsDCs, which lack cell surface expression of MHC class I molecules and cannot elicit MHC class I-dependent

**Fig. 4 | Primary DCs upregulate MHC class I and increase lipid peroxidation upon antigen presentation ex vivo and in vivo during immunization. a** Ex vivo experimental setup for B-D. **b** Flow cytometry analysis of biotin staining of splenic DCs pulsed with the indicated peptide (LCMV GP$_{61-80}$ or OVA$_{323-339}$). No biotin-LPETG indicates pulsing with OVA$_{323-339}$ but no addition of biotin-LPETG to the coculture. Flow cytometry analysis of the geometric mean (GeoMean) of (**c**) H-2K$^b$ or (**d**) lipid peroxidation in the indicated population of DCs. **e** In vivo experimental setup for F-H. **f** Flow cytometry analysis of biotin staining of mDCs extracted from popliteal lymph nodes (pLNs) from draining footpads immunized with the indicated formulation. **g** Flow cytometry analysis of the geometric mean (GeoMean) of H-2K$^b$ in the indicated population of migratory DCs (mDCs) from pLNs. **h** Flow cytometry analysis of sorted mDCs from immunized pLNs after staining with the BODIPY 581/591 C11 lipid peroxidation probe. Data show the GeoMean in the B530/530 channel gated in CD11c$^+$MHCII$^+$ from the indicated population of sorted mDCs. For (**b**, **c**, **f**) n = 6, (**d**) n = 5, (**g**) n = 7, and (**h**) n = 6 (PBS:Alum) or = 11 (OVA:Alum). Replicates are biological (**b–d**) or animals (**f–h**). Bar plots indicate mean ± SEM. Statistical tests are two-tailed unpaired t-tests between different sample conditions (**f**) and paired between biotin$^+$ and biotin$^-$ from same samples (**g**, **h**) or one-way ANOVA (**c**, **d**, **b**). All data is representative of at least two independent experiments. p-values are indicated, with ns for p-value > 0.05.

CD8$^+$ specific T cell responses[42]. Interestingly, survival differences induced by psDCs and nsDCs vanished (Fig. 5g, h). These results indicated that psDC transfer can protect mice from a *Listeria monocytogenes* challenge by directly mounting protective CD8$^+$ T cell responses via presentation on their own MHC class I molecules.

### psDC transfer promotes pathogen-specific CD8$^+$ effector and memory responses during influenza virus infection

We next determined whether the transfer of psDCs could also induce a higher response of pathogen-specific CD8$^+$ T cells against viral infection. To do so, mice that had been previously inoculated intranasally with psDCs or nsDCs were infected with influenza A virus (IAV) and the circulating T cell response during acute infection was evaluated (Fig. 6a and Supplementary Fig. 5a). Interestingly, we observed an increased percentage of CD8$^+$ T cells in the T cell compartment 9 days after infection when mice had been inoculated with psDCs compared to nsDCs, whilst 5 days after infection this difference was not evident (Fig. 6b). Moreover, there was an increase in the proportion of IAV-specific CD8$^+$ T cells in the psDC group (Fig. 6c). This promotion of IAV-specific CD8$^+$ T cells by psDCs was significant compared to PBS treatment, which did not differ from nsDCs treatment (Supplementary Fig. 5b, c). When looking at specific subsets, we observed increased levels of CD44$^+$CD62L$^-$ (Fig. 6d) but not CD44$^+$CD62L$^+$ CD8$^+$ T cells 9 days after infection (Supplementary Fig. 5d). These accounted for the increased proportion of CD8$^+$ T cells in the T cell compartment 9 days after infection. Indeed, the effector CD44$^+$CD62L$^-$ CD8$^+$ T cells displayed a greater proportion of IAV-specific cells in the psDC group (Fig. 6e), which did not appear in CD44$^+$CD62L$^+$ CD8$^+$ T cells (Supplementary Fig. 5e). Hence, psDCs were enhancing acute IAV-specific CD8$^+$ T cell responses with an effector phenotype compared to nsDCs. On the contrary, the proportion of CD4$^+$ T cell response in the T cell population did not vary at any time point evaluated (Supplementary Fig. 5f). No differences were observed when we evaluated the CD44$^+$CD62L$^-$ or the CD44$^+$CD62L$^+$ compartments separately (Supplementary Fig. 5g, h). Since intranasally-administered DCs were reaching the lung and not the mediastinal lymph nodes (Supplementary Fig. 4f), we assessed whether psDCs were directly responsible for the activation of IAV-specific CD8$^+$ T cells. Hence, we transferred psDCs or PBS to Batf3$^{-/-}$ mice, which lack cross-presenting cDC1s and display impaired influenza-specific CD8$^+$ T cell responses[43]. Following IAV infection, mice that had been transferred psDCs generated a greater proportion of IAV-specific CD8$^+$ T cells in peripheral blood 9 days after infection (Supplementary Fig. 5i), indicating that psDCs were directly responsible for generating antigen-specific CD8$^+$ T responses in this model.

Furthermore, we addressed whether the differential virus-specific CD8$^+$ T cell response during acute infection shaped the generation of memory T cell subsets. To do so, we evaluated the memory T cell response in the spleen of mice infected with IAV that had been previously inoculated with psDCs or nsDCs (Fig. 6a and Supplementary Fig. 5j). Interestingly, we observed a reduction in the proportion of central memory CD8$^+$ T cells (Tcm, CD44$^+$CD62L$^+$) in the CD8$^+$ T cell compartment (Fig. 6f), whereas there was a greater proportion of IAV-specific cells within Tcm (Fig. 6g). Likewise, the proportion of effector memory CD8$^+$ T cells (Tem, CD44$^+$CD62L$^-$) in the psDC group was higher in the CD8$^+$ T cell population (Supplementary Fig. 5k), although there were no differences in the presence of IAV-specific cells within the Tem compartment (Supplementary Fig. 5l). The proportion of Tcm in CD4$^+$ T cells remained unchanged (Supplementary Fig. 5m) whereas the proportion of Tem was reduced (Supplementary Fig. 5n). A greater proportion of IAV-specific CD8$^+$ T cells in the psDC group was also observed when splenocytes were restimulated with IAV peptides NP$_{366-374}$ and PA$_{224-233}$, as psDC treatment increased the proportion of IFNγ-secreting CD8$^+$ T cells in the global CD8$^+$ T cell population (Fig. 6h). This effect was due to the stimulation of a greater proportion of Tcm, but not Tem CD8$^+$ T cells (Fig. 6i, j). Hence, the inoculation of psDCs promotes effector pathogen-specific CD8$^+$ T cell response during acute infection, which translated into the generation of a greater proportion of IAV-specific CD8$^+$ T cells in the Tcm population, while increasing the proportion of Tem cells.

### Depletion of interacting DC:T cells during OVA:Alum immunization in vivo abrogates antigen-specific CD8$^+$ T cell responses

To assess whether the endogenous psDCs were responsible for the generation of CD8$^+$ T cell responses in an in vivo physiological context, we used again the LIPSTIC mouse model. As psDCs are biotinylated upon immunization, we used a streptavidin conjugated to saporin (SA-Saporin), an irreversible ribosome-blocking protein, to deplete interacting DC:T cells during immunization. We immunized *Cd40$^{GS}$* mice that had received OT-II CD4$^+$ T cells from CD4-Cre *Cd40lg$^{Srta}$* mice in the footpad with OVA:Alum. 72 h after immunization, interacting cells were biotinylated in vivo by injecting biotin-LPETG or PBS. Before the surface biotin is recycled[23], SA-Saporin was injected to deplete biotin$^+$ mDCs (psDCs) (Fig. 7a). Then, we analyzed the antigen-specific CD8$^+$ T cell response in the draining lymph nodes 8 days after immunization by tetramer staining (Supplementary Fig. 6a). As expected, SIINFEKL-specific CD8$^+$ T cells were only detected in the draining pLNs from the immunized footpads but not the contralateral ones. Strikingly, when interacting cells had been biotinylated, SA-Saporin administration resulted in a significant 50% reduction in the number of SIINFEKL-specific CD8$^+$ T cells (Fig. 7b). The reduction in antigen-specific CD8$^+$ T cells was more prominent in the CD44$^+$CD62L$^-$ than in the CD44$^+$CD62L$^+$ compartment, indicating a greater reduction in antigen-specific CD8$^+$ T cells with an effector phenotype (Fig. 7c, d). Furthermore, SA-Saporin injection upon cell biotinylation also reduced the number of total CD8$^+$CD44$^+$ T cells (Fig. 7e). This indicates that depletion of interacting DC:T cell pairs abrogates the generation, not only of SIINFEKL-specific CD8$^+$ T cells, but may also affect other OVA-specific CD8$^+$ T cells. Additionally, we cannot rule out the activation of bystander CD8$^+$ T cells by other means such as cytokine secretion[44]. Interacting OT-II *Cd40lg$^{Srta}$* CD4$^+$ T cells are also biotinylated in this system. However, the total size of the population was not significantly reduced 8 days after immunization (Supplementary Fig. 6b), which may indicate that the observed effects stem from the depletion of the other biotinylated partner of the interaction: psDCs. Hence, we found that psDCs are partly responsible for the generation of antigen-specific CD8$^+$ T cells responses during immunization in vivo.

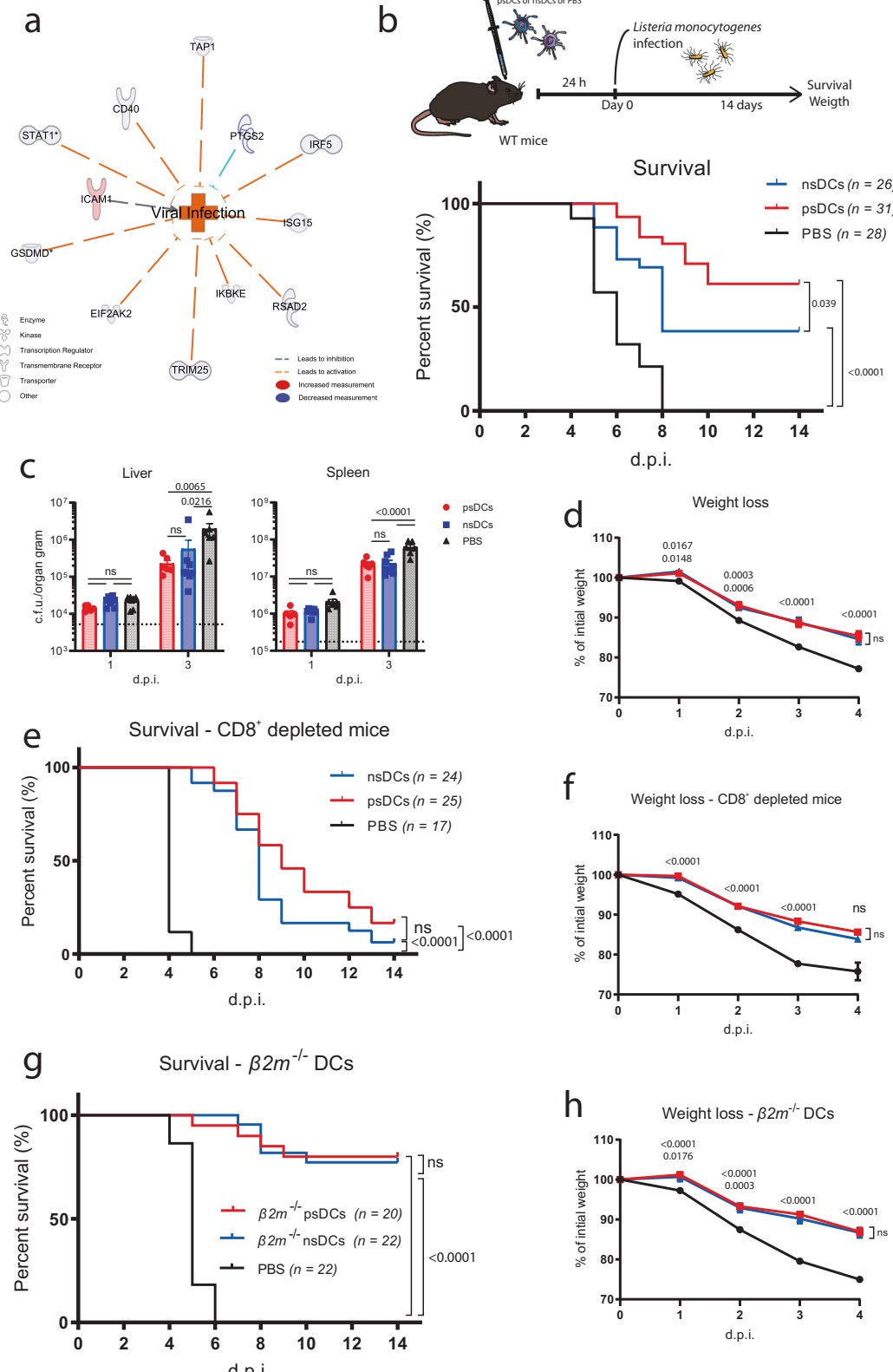

## Discussion

DCs receive help from CD4+ T cells to activate CD8+ T cell responses[2,4,45,46]. Apart from the mechanistic determination that licensing is mediated by CD40 induction[5–7], to our knowledge no other studies have addressed the cellular and molecular mechanisms that enable DCs licensing. This is because one of the major challenges of studying DC licensing is to identify the DCs that become licensed. In

this study, we used post-synaptic BMDCs[20] and the LIPSTIC system[23] to characterize the cellular and molecular mechanisms governing DC licensing.

The transcriptomic reprogramming observed in psDCs[20] includes many genes induced by the transfer of extracellular vesicles during antigen presentation[19], while others may rely on the induction of CD40 or other receptor-ligand interactions taking place during

**Fig. 5 | Transfer of psDCs protects mice from *Listeria monocytogenes* infection via MHC class I-mediated CD8⁺ T cell activation. a** IPA protein network representing the prediction for the functional term *Viral infection* for the proteomics data. **b** Experimental design followed in infection experiments (up) and survival curve of mice infected with *L. monocytogenes* after psDCs, nsDCs or PBS transfer. **c** Bacterial load in the liver and spleen at 1 or 3 days after infection in mice that had been transferred psDCs, nsDCs or PBS. Data are representative of two independent experiments, with *n* = 5 and 6 animals for psDC day 1 and 3, respectively; *n* = 8 for nsDC and *n* = 6 and 7 for PBS day 1 and 3, respectively. Dashed line indicates limit of detection. **d** Evolution of weight as proportion of initial weight of mice in (**b**), with *n* and color code as in (**b**). **e** Survival curves and (**f**) percentage of initial weight of

mice injected with an anti-CD8 antibody for CD8⁺ T cell depletion previously to psDCs, nsDCs or PBS transfer and then infected with *Listeria monocytogenes* (*n* and color code as in (**e**)). **g** Survival curves and (**h**) evolution of weight (*n* and color code as in (**g**)) of mice infected with a lethal dose of *Listeria monocytogenes* that had been transferred *β2m⁻/⁻* psDCs, *β2m⁻/⁻* nsDCs or PBS. In (**b** and **d**) data shown are a pool of 3 independent experiments. In (**e–g**), data was pooled from two independent experiments. Bacterial load was analyzed using unpaired two-way ANOVA. Statistical analysis of survival curves was performed with the Gehan-Breslow-Wilcoxon test. Weights were compared using a mixed-effect model with the Geisser–Greenhouse correction and Tukey's multiple comparison test. (**c**, **d**, **f**, and **h**) indicate mean ± SEM. D.p.i.: days post-infection. ns: *p*-value > *0.05*.

immune synapse formation. Our data show that psDCs display a distinct proteome signature. Remarkably, MHC class I molecules are more abundant in psDCs, together with proteins from other routes related to cross-presentation and MHC-I-mediated CD8⁺ T cell priming. Accordingly, psDCs displayed enhanced cross-presentation of soluble OVA protein. The upregulation of these processes explains the induction of cross-presentation by CD4⁺ T cell help[1]. In the case of the observed increase of MHC-I proteins, a previous study reported that CD40 activation using a monoclonal antibody induced MHC-I[47]. Our results show that the engagement of CD40 during synaptic interactions promotes MHC-I upregulation as blocking CD40-CD40L interaction decreased MHC-I levels in both psDCs and nsDCs. However, other mechanisms may underlie MHC-I upregulation during licensing as the differences between psDCs and nsDCs did not disappear. On the other hand, the increased lipid peroxidation in DCs is induced by innate stimulation[36]. However, our results are consistent with synaptic contacts inducing lipid peroxidation both in vitro and in vivo during immunization. As genes upregulated in psDCs are mostly related to DC maturation and activation[20], synaptic contacts may be inducing lipid peroxidation via amplification of DC activation pathways on psDCs. Increased lipid peroxidation may be explained by an increment in ROS production induced by innate signals[36]. However, alterations of the peroxidation homeostasis would also increase lipid peroxidation. Indeed, peroxiredoxins, which remove hydrogen peroxide and alkyl hydroperoxides, were reduced in psDCs. Lipid peroxidation is a potential mechanism that affects antigen export to the cytosol during cross-presentation[31,36,48]. Our data demonstrate that prevention of lipid peroxidation with α-tocopherol abrogates cross-presentation by psDCs. Moreover, synaptic interactions seem to mediate the increased lipid peroxidation in psDCs as blocking CD40-CD40L interactions during synaptic contacts decreased lipid peroxidation in psDCs. Also, blocking CD40L made psDC display lower levels of lipid peroxidation than nsDCs, existing no differences between them and nsDCs generated in the presence of an isotype control. Interestingly, nsDCs generated in the presence of anti-CD40L had increased levels of lipid peroxidation, possibly indicating that CD40 can induce, but may also have a role in controlling lipid peroxidation levels within a range to prevent excessive lipid peroxidation, which merits further investigation.

Antigen presentation is a key event in the generation of adaptive immune responses, which are the primary goal of immunization. Vaccination mobilizes DCs to draining lymph nodes where they remain functional for long periods of time[49]. This is probably due to survival signals DCs receive during cognate synaptic interactions[24]. Indeed, after immunization, DCs can acquire enhanced immune functions such as an innate memory-like phenotype[25]. Previous studies characterized the total DC population from immunized lymph nodes where cognate interactions had taken place[21]. However, no observed effect could be attributed specifically to the interacting DC population, but rather associated to the total DC population during an ongoing immune response. Interestingly, this analysis predicted

upregulation of MHC-I presentation routes in DCs of challenged lymph nodes at the gene expression level[21]. Thanks to the LIPSTIC model, we have addressed this issue in the context of footpad vaccination with a soluble antigen formulated in alum, the most widely used adjuvant for human vaccines. The biotinylation of psDCs was used to deplete them in vivo. SA-Saporin has been successfully used in combination with biotinylated antibodies, tetramers or other molecular ligands to deplete specific cell populations in vivo by inducing cell death in the target cells[50,51]. This system has allowed us to demonstrate that interacting DC:T cell pairs at 72 after immunization, when the peak of interactions occurs[23], are key for generating antigen-specific CD8⁺ T cell responses via cross-presentation of a soluble antigen. As the number of OT-II cells was not significantly reduced, the effect may be due to depletion of psDCs. Indeed, depletion of CD4⁺ T cells later than 3 days after antigen administration does not affect the CD8⁺ T cell memory response generation in different experimental models[10,12,15]. Interestingly, cDC2s are the most prominent interacting population in this system[23]. However, cDC1s were described to receive CD4⁺ T cell help for cross-presentation of the transferred antigen in vitro[4] and in vivo upon infection[52,53] and in anti-tumor responses[3]. This could be due to the previous unavailability of reliable cDC2-deficient models[54], or to the fact that the licensed population depends on the context. Furthermore, alum is a Th2-skewed adjuvant that could preferentially mobilize cDC2s.

In support of the observation that psDCs optimize CD8⁺ T cell responses, we showed that psDCs can be used as a form of immune intervention against infectious diseases, decreasing mortality induced by *Listeria*. Although licensing was not described as relevant for CD8⁺ T cells in this model[10], the protective effect of psDCs was CD8⁺ T cell-dependent and directly exerted by psDCs. Surprisingly, administration of β2m⁻/⁻ DCs further increased protection from infection. This however may be due to (i) the transfer of cells without MHC class I molecules activating NK cells that aid in the protection, (ii) the reduced ability to generate an adaptive response translating into a reduced immunopathology, or (iii) the great variability in death rates observed in different experiments despite using the same bacterial dose. Moreover, we also show that the administration of psDCs augmented pathogen-specific CD8⁺ T cell responses during the acute and the memory phase of influenza A virus infection. Transferred DCs do not reach draining mediastinal lymph nodes, but rather stay in the lung, which raises the question of whether psDC can directly activate IAV-specific CD8⁺ T cell there. Previously, it has been reported that naïve T cells can be found in lung parenchyma[55] and migrate from lung to lymph nodes[56]. Moreover, primary activation of CD8⁺ T cells in non-secondary lymphoid organs has been shown[57]. Indeed, our data show that licensed psDCs are directly responsible for the generation of influenza-specific CD8⁺ T cells, as transferred psDCs into Batf3⁻/⁻ mice, which lack cross-presenting cDC1s and display impaired CD8⁺ T cell responses during influenza infection[43], increase IAV-specific CD8⁺ T cell levels. Pathogen-specific dependency on licensing for generating CD8⁺ T cells has been previously described[4,9] and may be due to

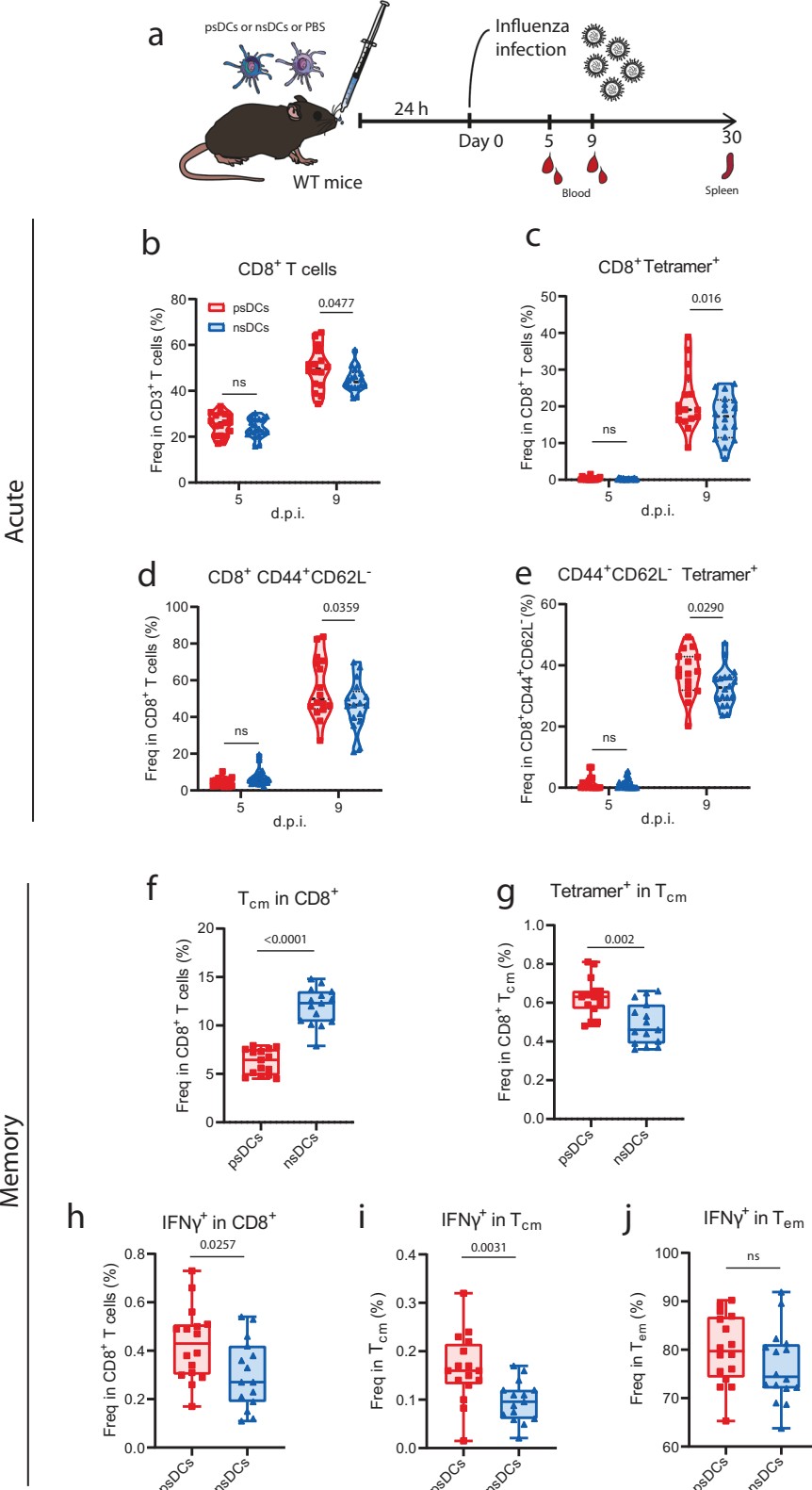

distinct innate signals bypassing the dependency of synapse-mediated training[11]. Hence, their therapeutic potential may depend on the pathogen targeted.

Collectively, our data sustain a further rationale for the improvement of vaccination technologies and immunotherapy for infection and cancer that may be relatively independent of the specific antigen used to drive initial activation.

## Methods

### Mice

Mouse strains included in this study comprise C57BL/6JOlaHsd (or C57BL/6) wild-type mice; B6-SJL (Ptprc[a] Pepc[b]/BoyJ) expressing CD45.1 allele; TCR transgenic OT-II mice (B6.Cg-Tg(TcraTcrb) 425Cbn/J) and TCR transgenic OT-I mice (C57BL/6-Tg(TcraTcrb) 1100Mjb/J) both mated with B6-SJL (Ptprc[a] Pepc[b]/BoyJ); *Rag1*[-/-] mice

**Fig. 6 | Transfer of psDCs increases the pathogen-specific acute and memory CD8⁺ T cell response during influenza virus infection. a** Experimental procedure performed for IAV infection upon psDCs/nsDCs transfer. FACS analysis in peripheral blood during acute infection at day 5 and 9 after infection showing: the frequency of (**b**) CD8⁺ T cells within CD3⁺ T cells, (**c**) tetramer⁺ cells within CD8⁺ T cells, (**d**) CD44⁺CD62L⁻ cells within CD8⁺ T cells, (**e**) tetramer⁺ cells within the CD44⁺CD62L⁻ CD8⁺ T cells. Color code for (**c–e**) as in (**b**). FACS analysis in spleen of mice during the memory phase day 30 after infection showing: the frequency of (**f**) Tcm (CD44⁺CD62L⁺) within CD8⁺ T cells, and (**g**) tetramer⁺ cells within Tcm CD8⁺ T cells. FACS analysis of splenocytes collected on day 30 after infection and

restimulated in vitro with the influenza virus NP$_{366\text{-}374}$ and PA$_{224\text{-}233}$ peptides showing: proportions of (**h**) IFNγ⁺ cells within CD8⁺ T cells, (**i**) IFNγ⁺ cells within Tcm CD8⁺ T cells, and (**j**) IFNγ⁺ cells within Tem (CD44⁺CD62L⁻) CD8⁺ T cells. All data are representative of two independent experiments. For (**b–e**) $n = 18$ for nsDC and $n = 17$ at day 5 and $n = 16$ at day 9 for psDC, while for (**f**, **g**) $n = 15$ and for (**h–j**) $n = 16$. Statistical analyses are result of a mixed-effect model with and Sidák's multiple comparison test (**b–e**) or two-tailed unpaired $t$-test (**f–j**). $p$-values are indicated, with ns for $p$-value $> 0.05$. MFI: mean fluorescence intensity. d.p.i.: days post-infection.

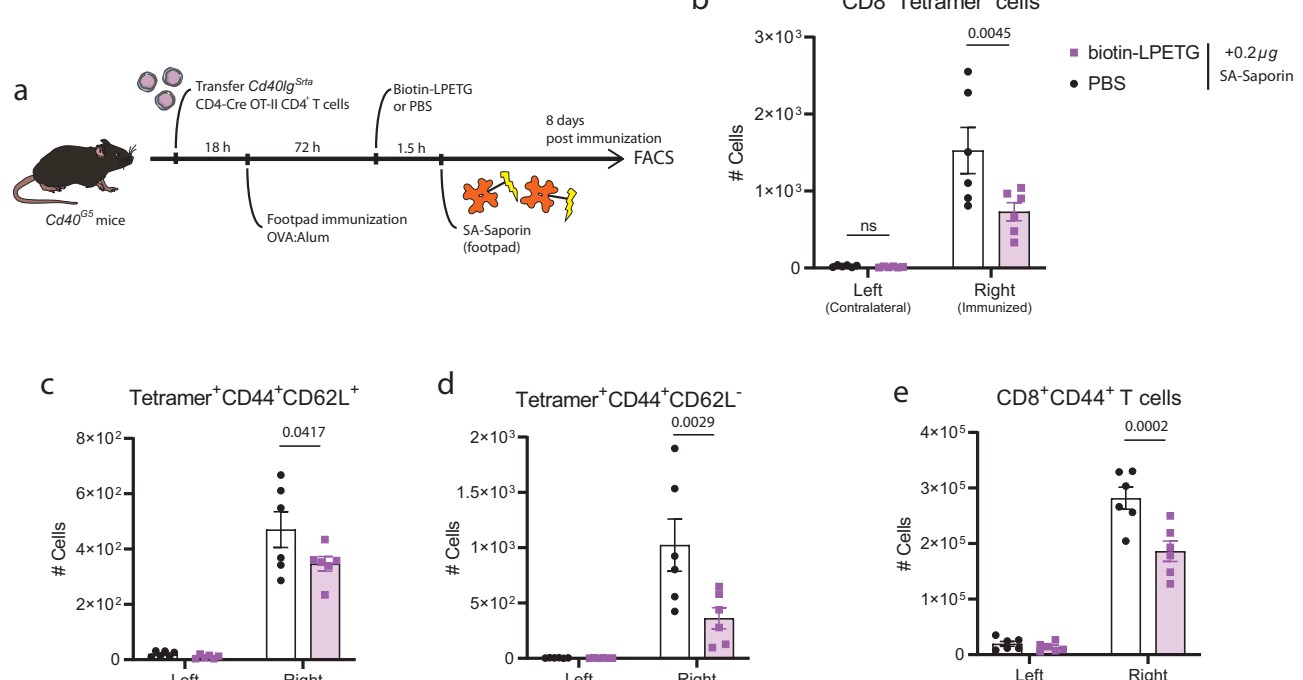

**Fig. 7 | Depletion of psDCs generated during immunization abrogates antigen-specific CD8⁺ T cell responses. a** Experimental set up for evaluation of CD8⁺ T cell responses upon psDC depletion. **b** Count of SIINFEKL-specific CD8⁺ T cells in draining popliteal lymph nodes from the immunized (right) or contralateral (left) side obtained by H-2K$^b$-OVA$_{257\text{-}264}$ tetramer staining in flow cytometry. Count of (**c**)

Tetramer⁺CD44⁺CD62L⁺ and (**d**) Tetramer⁺CD44⁺CD62L⁺ CD8⁺ T cells. **e** Count of the total CD8⁺CD44⁺ population. The count represents the cells contained in one full pLN. $N = 6$ animals. Color code for (**c–e**) as in (**b**). Bar plots indicate mean ± SEM. Data are representative of two independent experiments. Statistical tests are unpaired two-way ANOVA. $p$-values are indicated, with ns for $p$-value $> 0.05$.

(B6.129S7-Rag1tm1Mom/J); $\beta 2m^{-/-}$ mice (B6.129P2-B2m$^{tm1Unc}$/DcrJ) from The Jackson Laboratories. $Cd40^{G5}$ and $Cd40lg^{Srta}$ (LIPSTIC mice) were kindly provided by Dr. Giulia Pasqual (University of Padova, Italy). Batf3-/- mice (B6.129 S(C)-Batf3tm1Kmm/J) were kindly provided by Dr. David Sancho (CNIC). Mice were kept on dark/light cycle 12:12, ambient temperature 22 °C ± 2, and humidity 55% ± 10. Female 8–12-week-old mice were used unless otherwise indicated. In the case of LIPSTIC mice, both males and females were used. Littermates were randomly assigned to experimental groups. All mice were euthanized using a CO$_2$ chamber. Animal experiments were approved by the local Ethics Committee for Basic research at the CNIC Ethical Committee for Animal Welfare and the Organo Encargado del Bienestar Animal (OEBA) del Gabinete Veterinario de la Universidad Autonoma de Madrid (UAM) and are in agreement with EU Directive 86/609/EEC and Recommendation 2007/526/EC regarding the protection of animals used for experimental and other scientific purposes, enforced in Spanish law under Real Decreto 53/2013 (Authorization ProEX 206.1/20). Additionally, experiments were approved by Italian Ministry of Health (Authorization n. 994/2020-PR). All animals were housed and/or bred in the pathogen-free animal facility of the Centro Nacional de Investigaciones Cardiovasculares Carlos III (Madrid) or Università degli Studi Padova in accordance with the animal care

standards of the institutions. Food and water was provided *ad libitum*. Experimental procedures were designed and performed taking into account the ARRIVE guidelines.

### Cell culture and psDC generation

All mouse cells were cultured in RPMI 1640 (Gibco) supplemented with 10% fetal bovine serum (FBS, Sigma Aldrich), penicillin (100 U/ml) and streptomycin (100 μg/ml, Sigma Aldrich), HEPES (20 mM pH 7.5, Gibco), L-Glutamine (2 mM, Sigma Aldrich), and β-mercaptoethanol (50 μM, Merck). BMDCs for the generation of psDCs were generated similarly to previously reported[20]. Briefly, erythrocyte-depleted bone marrows were cultured on non-treated 150 mm Petri dishes in the presence of GM-CSF (20 ng/ml). On day 3, cells in suspension were collected and plated with fresh medium with GM-CSF. On day 6, cells were detached with phosphate-buffered saline (PBS), EDTA (5 mM), and bovine serum albumin (BSA) 0.5% (PBE) and plated with fresh medium with GM-CSF. At days 9 or 10 DCs were plated to 2.5·10⁶ cells/well·mL in non-adherent 6-well plates, and matured with LPS (250 ng/ml) for 6 h. In the last 1.5 h, the OVA$_{323–339}$ peptide was added or not to pulse DCs at 5 μg/ml. OT-II CD4⁺ T cells were purified from spleen and lymph nodes using the EasySep Mouse CD4⁺ T Cell Isolation Kit (Cat #19852, STEMCELL Technologies) following manufacturer's instructions plus an addition of

biotinylated anti-CD25 antibody (7D4, BD Biosciences) to the isolation antibody cocktail. OT-II CD4$^+$ T cells were added at a 1:2 (DC:T cell) ratio and were cocultured overnight. After coculture, DCs were purified by negative selection using the EasySep Mouse CD90.2 Positive Selection Kit II (Cat #18951, STEMCELL Technologies). In the case of the use of activated CD4$^+$ T cell, CD4$^+$ T cells from peptide-pulsed cocultures were purified after overnight coculture using the EasySep Mouse CD4$^+$ T Cell Isolation Kit. In flow cytometry analysis of BMDCs, DCs were identified as CD11c$^+$ cells.

## Confocal microscopy

DCs were adhered to fibronectin-coated glass-bottom chambers (80827, ibidi GmbH) and fixed with 2% paraformaldehyde in PHEM (PIPES 30 mM, Hepes 20 mM, EDTA 2 mM, MgCl2 1 mM, pH: 6.9) containing 0.12 M sucrose for 15 min at R/T. Cells were then permeabilized with the same solution containing Triton-X100 (0.2%) and treated with Fc-block (anti-CD16/CD32) in blocking buffer containing 3% BSA and human γ-globulin 50 µg/mL in PHEM for 30 min at RT. Mouse monoclonal anti-α-Tubulin conjugated to FITC (clone DM1A, F2168 Sigma Aldrich) and Alexa Fluor 647-conjugated Phalloidin (A22287 ThermoFisher Scientific) were incubated in the same solution for 2 h at RT. Chambers were washed in Tris-buffered saline (pH: 7.4); upon completion of staining, cells were imbibed in Prolong Gold mountant medium with DNA stain DAPI (P36931 ThermoFisher Scientific). A series of fluorescence and brightfield frames were captured using a Leica SP8 Navigator Confocal Microscope equipped with a pulsed white light laser (WLL, range 470-670 nm) and an HC PL Apo CS2 100x/1.4 OIL objective. Images were acquired at room temperature (25 °C) with hybrid detectors and processed with the accompanying Application Suite X software (LAS X, 3.5.2. 18963; Leica Microsystems GmbH) and Image J software (http://rsbweb.nih.gov/ij/).

## Proteomics

Protein extracts were obtained from cell pellets using Tris-HCl 100 mM pH6.8 with 10 mM DTT, 2% SDS and boiled 5 min. Proteins were on-filter digested using the FASP technology (Expedeon) and sequencing grade trypsin (Promega) in a 1:40 (w/w) trypsin: protein ratio at 37 °C overnight. The resulting tryptic peptides were subjected to multiplexed isobaric labeling (TMT; Thermo Fisher Scientific, Bremen, Germany) according to the manufacturer's protocol. The differentially tagged samples were then pooled, desalted on Waters Oasis HLB C18 cartridges (Waters Corporation, Milford, MA, USA) and dried-down. Labeled peptide samples were analyzed by liquid chromatography tandem mass spectrometry (LC-MS/MS) using an Ultimate 3000 HPLC system (Thermo Fisher Scientific) coupled via a nanoelectrospray ion source (Thermo Fisher Scientific) to a Q Exactive HF mass spectrometer (Thermo Fisher Scientific). C18-based reverse phase separation was performed using a PepMap 100 C18 5 µm 0.3 × 5 mm as trapping column (Thermo Fisher Scientific) and a PepMap RSLC C18 EASY-Spray column 50 cm × 75 µm ID as analytical column (Thermo Fisher Scientific). Peptides were loaded in buffer A (0.1% formic acid in water (v/v)) and eluted with a linear gradient consisting of 0-21% buffer B (100% acetonitrile, 0.1% formic acid (v/v)) for 300 min and 21–90% B for 5 min at a flow rate of 200 nl/min. MS spectra were acquired in the Orbitrap analyser using full ion-scan mode with a 390-1500 m/z range and 120,000 FT resolution. MS/MS was performed in a data-dependent manner using the top-15 adquisition mode. HCD fragmentation was performed at 30% of normalized collision energy and MS/MS spectra were analysed at 30,000 resolution in the Orbitrap.

Protein identification was performed using the SEQUEST HT algorithm integrated in the Proteome Discoverer 2.1 (Thermo Fisher Scientific). MS/MS scans were matched against a concatenated protein database containing mouse sequences (September 2020 release) and the corresponding inverted sequences. For database searching, parameters were as follows: trypsin digestion with a maximum of two missed cleavage sites; 800 ppm and 0.02 Da precursor and fragment mass tolerance, respectively. TMT modification at N-terminus and Lys and carbamidomethylation at Cys were selected as fixed modifications, whereas oxidation at Met, Pro, Trp, His, and Asp, dioxidation at Met, Pro and Trp, and trioxidation at Trp were set as variable modifications. The false discovery rate (FDR) for peptide identification was calculated using the probability ratio method after a 15 ppm precursor mass tolerance postfiltering[58,59]. A 1% FDR criterion was used to ascertain true identification.

Quantitative information was extracted from the intensity of TMT reporter ions. Analysis of protein abundance was performed using the SanXoT package[60] based on the Weighted Scan, Peptide and Protein (WSPP) statistical model[61], which uses raw quantifications as input data and computes the protein log2-fold changes for each sample with respect to the average value of all the nsDC samples. Protein log2-ratios were expressed in units of standard deviation according to their estimated variances (Zq values). Averaged quantitative values for each protein were obtained applying the Generic Integration Algorithm and the final[27] quantitative data was calculated by the ratio of psDCs vs. nsDCs. For the analysis of coordinated protein changes we used the Systems Biology Triangle (SBT) statistical model[27], which estimates standardized functional category changes (Zc). Proteins were functionally annotated using Gene Ontology, KEGG, Panther, Reactome, and CORUM databases. A 5% FDR criterion was used to asses significant abundance changes, both at Zq and Zc levels. Quantitative abundance changes at peptide level were determined by the standardized variable (Zpq), expressing the deviation between the log2-ratio quantifications of each peptide from the protein they come from. Partial least squares discriminant analysis (PLS-DA) was calculated and represented using the MetaboAnalyst 5.0 software (http://www.metaboanalyst.ca). All identified proteins and their corresponding Zq values were used as source data for the Ingenuity Pathway Analysis software (QIAGEN).

## Flow cytometry and sorting

Flow cytometry samples were first stained for cell viability according to manufacturer's instructions and Fc receptors were blocked using Anti-mouse CD16/CD32 (Fc Shield) for 15 min in PBS at 4 °C. Then, samples were resuspended in antibody cocktails for a minimum of 20 min in ice-cold PBE. For intracellular staining, cells were then fixed, permeabilized and stained with the BD Cytofix/Cytoperm kit (Cat# 554714, BD Biosciences) following manufacturer's instructions. Antibodies and tetramers were used at a 1/200 dilution unless stated otherwise. Samples were acquired using an LSR Fortessa or in BD FACSymphony (BD Biosciences) and analyzed with FlowJo software version 10 (TreeStar). To sort biotin$^+$ and biotin$^-$ mDC populations, cell suspensions from pLNs (see LIPSTIC in vivo section) were first stained with DAPI (2.5 µg/mL) and Fc Shield for 15 min at 4 °C and then stained for surface markers to be sorted within the Live CD11c$^+$MCHII$^{hi}$ population using a BD FACSAria Fusion cell sorter. Sorted cells were resuspended at the same concentration and then stained for lipid peroxidation (see lipid peroxidation section).

## Cross-presentation assays in vitro

For cross-presentation assays, EndoFit Ovalbumin (Invivogen) protein was added at 500 µg/ml or the indicated concentration together with LPS (250 ng/mL) for 6 h to BMDCs. In the last 1.5 h, 5 µg/ml of the OVA$_{323–339}$ peptide was added to the psDC sample. Then, OT-II CD4$^+$ T cells or media were added to the psDC or nsDC sample respectively at a 1:2 (DC:CD4) ratio and were cocultured overnight. To prevent immune synapse in the non-pulsed fraction due to the presence of the OVA protein, no CD4$^+$ T cells were added in the non-pulsed fraction. DCs were then purified and incubated with effector or naïve OT-I CD8$^+$ T cells at a 5:1 (DC:CD8) ratio in U-bottom 96-well plates, using $10^5$ psDCs or nsDCs per well. To generate effector OT-I cells, splenocytes from OT-I mice were activated with 0.1 µM SIINFEKL

and 100 ng/ml IL-2 in complete RPMI for 2 days, and rested in complete medium with IL-2 for another 7 days. Effector CD8+ T cells were cocultured for 2 h and then Brefeldin A (5 mg/ml) was added to the medium and incubated another 4 h until cells were stained for surface markers and intracellular IL-2 as indicated. In the case of naïve CD8+ T cells, OT-I cells were purified from spleens of OT-I mice with the EasySep Mouse CD8+ T Cell Isolation Kit (Cat# 19853, STEMCELL Technologies), stained with CellTrace Violet Cell Proliferation Kit (Cat# C34557, Thermo Fisher Scientific) according to the manufacturer's instructions and then cocultured for 72 h. Proliferation was analyzed using the proliferation analysis tool included in the FlowJo software. For inhibition of lipid peroxidation in cross-presentation assays, the indicated concentrations of α-tocopherol diluted in ethanol or the same volume of ethanol were added to the coculture at the same time that CD4+ T cells were added.

## Lipid droplet and lipid peroxidation

To determine lipid peroxidation or quantify lipid bodies, cells were counted and resuspended at equal concentrations in BODIPY 581/591 C11 or BODIPY 493/503 respectively diluted to 0.8 µM in PBS and incubated for 35 min at 37 °C. Cells were then washed with ice-cold PBE and viability (LIVE/DEAD Fixable Blue or Zombie Aqua Fixable Viability Kit for ex vivo experiments) and surface marker stating was performed for 20 min at 4 °C, washed and acquired. Lipid peroxidation or neutral lipid content was determined within CD11c+ (BV421) cells for BMDCs as the mean fluorescence intensity (MFI) in the B530/30 emission channel, excited with the 488 nm laser. For lipid peroxidation in vivo, sorted biotin+ or biotin- were resuspended at the same concentration, stained for lipid peroxidation, and later acquired in flow cytometry.

## Seahorse XF glycolytic assay

Cell respiration was measured with a Seahorse XF96 extracellular flux analyzer (Agilent Biosciences). Cells were plated at $2 \cdot 10^5$ cells/well in poly-L-Lys precoated XF96 FluxPak plates (Agilent Technologies). Injections of glucose (10 mM), oligomycin (1.5 µM), and 2-deoxy-D-glucose (50 mM) were performed. Three mix and measure steps of 3 min each were performed for resting conditions and following each injection. Plates included 8 biological replicates and 6 technical replicates.

## LIPSTIC ex vivo

Ex vivo antigen presentation assays with LIPSTIC primary cells were performed as previously described[23]. Briefly, spleens were perfused with Liberase TL (250 µg/mL) and DNAse I (10 µg/mL) and incubated at 37 °C for 20 min in HBSS (Lonza). Spleens were then grinded on a 70 µm mesh and then erythrocytes lysed with ACK buffer (Lonza). DCs were isolated from those single-cell suspension by magnetic cell separation using CD11c MicroBeads Ultrapure mouse (Cat# 554714, Miltenyi Biotec) following manufacturer's instructions. DCs were then pulsed with LPS (10 µg/ml) and OVA$_{323-339}$ (10 µM) or LCMV GP$_{61-80}$ for 2.5 h at 37 °C in complete RPMI. Then, cells were washed three times and seeded into U-bottom 96-well plates with CD4+ T cells at a 1:1 ratio. Cells were cocultured for 6 h and in the last 20 min, 10 µM of biotin−LPETG was added to the media. Cells were washed three times with PBE before staining for flow cytometry. In this case, the live/dead probe used was Zombie Aqua Fixable Viability Kit (Cat# 423101, BioLegend). For biotin detection, an anti-biotin PE antibody (in MHC-I) or Streptavidin-BV421 (in lipid peroxidation) were used. Biotin−aminohexanoic acid−LPETGS (C-terminal amide, 95% purity) was purchased from GenScript (custom synthesis). In the case of experiments blocking CD40-CD40L interactions, an anti-CD40L (MR1 clone) or an isotype control (BioLegend) were added at 100 µg/mL from the beginning of coculture, which was cultured for 16 h before staining for flow cytometry.

## LIPSTIC in vivo and immunizations

LIPSTIC mice were used to analyze *in vivo*-occurring psDCs by using immunization experiments as previously described[23]. Briefly, $3 \cdot 10^5$ CD4+ T cells purified from spleens of male CD4-Cre OT-II *Cd40lg^{Srta}* mice were injected retro-orbitally into *Cd40^{GS}* mice. Then, 18 h after T cell transfer, mice were immunized with 15 µg OVA adsorbed to alum (2:1 volume ratio OVA:Alum) or PBS:Alum via subcutaneous injection in the hind footpad. Then, 72 h after immunization, 450 nmol of biotin-LPETG were injected subcutaneously into the hind footpad along six injections 20 min apart of 30 µl with 2.5 mM biotin-LPETG diluted in PBS. 40 min after the last injection popliteal lymph nodes were collected. Single-cell suspensions from popliteal lymph nodes were prepared by culturing lymph nodes Liberase TL (250 µg/mL) and DNAse I (10 µg/mL) for 20 min at 37 °C in HBSS, grinded on a 70 µm mesh and then stained for surface markers, including biotin with an anti-biotin PE antibody.

## DC transfer and infections

**Listeria monocytogenes.** We used *Listeria monocytogenes* wild type EGD (BUG 600) strain provided by Dr. Pascale Cossart (Pasteur Institute, Paris, France). After purification, psDCs or nsDCs were resuspended in PBS and $10^6$ cells transferred via retro-orbital intravenous (i.v.) injection. *Listeria monocytogenes* was grown for 36 h in a Brain Heart Infusion Broth (BHI) inverted plate. Then, a colony of was grown in BHI at 37 °C with shaking (250 rpm) overnight. To assess colony-forming units (CFUs) bacterial growth was measured by spectrophotometry and used in log-phase (optical density 0.2–0.8 at 600 nm). 24 h after DC transfer, $5.5 \cdot 10^5$ CFUs were injected i.v. in the lateral tail vein. For CD8+ T cell depletion, 24 h previous to DC transfer 50 µg of InVivoMAb anti-mouse CD8β (Lyt 3.2) were injected intraperitoneally (i.p.). Bacterial load in liver and spleen were determined at 1 and 3 days after infection. Briefly, organs were collected, weighted, and then tissue was mechanically disrupted using 7 mm of stainless-steel beads (Life Technologies, Grand Island, NY) in a TissueLyser LT (Qiagen, Hilden, Germany) 3-min cycle (40 oscillations/s) performed in PBS with Triton X100 0.1% to release intracellular bacteria. Organ homogenates were plated in BHI inverted plates, incubated for 36 h and then colonies were counted and the CFU per organ weight determined. Duration of CD8+ T cell depletion after a single injection was checked by flow cytometry in peripheral blood of injected or control mice. Weight loss and disease severity were monitored throughout infection.

**Influenza virus.** Influenza A/Puerto Rico/8/34 (PR8) virus and Influenza A/X-31(H3N2) (X31) were kindly provided by Dr. Estanislao Nistal-Villán (University CEU San Pablo, Madrid). After purification, psDCs or nsDCs were resuspended in PBS and $5 \cdot 10^5$ cells were administered intranasally in 40 µL by inhalation after briefly anesthetizing mice with inhaled isoflurane. 24 h after DC administration mice we infected intranasally with 40 µL of PBS containing 20 pfu of Influenza virus A/Puerto Rico/8/34 (PR8) virus, 300 pfu of Influenza virus A/X-31(H3N2) (X31) or PBS as indicated in the figures. For the experiment using Batf3$^{-/-}$ mice, 15 pfu of PR8 were administered.

## DC biodistribution

psDCs and nsDCs were generated from bone marrow of CD45.1+ mice. Cells were labeled with CSFE (2 µM, BioLegend) 5 min at 37 °C in PBS. Then, $5 \cdot 10^5$ or $1 \cdot 10^6$ CFSE-labeled cells were administered i.n. or i.v. respectively. 24 h after DC transfer, lungs, spleens and mediastinal lymph nodes were extracted and digested with liberase TL as previously stated. Then CD11c+ cells were purified from total organ homogenates with CD11c MicroBeads Ultrapure mouse (Miltenyi Biotec) following manufacturer's instructions. CD11c+ fractions were stained for flow cytometry and cell numbers of CD45.1+CD11c+CFSE+ cells were calculated by normalizing the values by multiplying by the ratio of added/read Trucount beads.

**Article** https://doi.org/10.1038/s41467-023-42480-3

## IAV-specific T cell responses

During acute infection, peripheral blood was extracted from the facial vein at days 5 and 9 after infection. Blood lymphocytes were separated using a Ficoll density gradient centrifugation. In the case of memory assays, spleens were collected 30 days after infection and splenocytes used. In both cases, cells were stained for viability and Fc receptors were blocked prior to surface staining with a mixture of antibodies and PE- or APC-labeled tetramers specific for H-2D$^b$ NP$_{366-374}$ (ASNEN-METM) and PA$_{224-233}$ (SSLENFRAYV). Tetramers were kindly provided by the NIH Tetramer Facility at Emory University. For analysis of splenocyte activation by measuring intracellular cytokines, splenocytes were re-stimulated to induce cytokine production by incubation of cell suspensions with an excess of NP$_{366-374}$ and PA$_{224-233}$ peptide (1 μM) for 6 h, brefeldin A (Sigma, 5 mg/ml) was added for the last 4 h of culture and cells were stained for flow cytometry.

## SA-saporin depletion and CD8$^+$ T cell responses

To study the effect of biotin$^+$ DCs (psDCs) depletion on the CD8$^+$ T cell response, we followed the LIPSTIC in vivo protocol. However, this time 450 nmol of biotin-LPETG or the same volume of PBS were injected in the footpad at 72 h after immunization. Then, 1.5 h later 0.2 μg of Streptavidin-Saporin (Cat# IT-27, ATS) were injected in the immunized footpad. Eight days after immunization, draining and contralateral popliteal lymph nodes were collected, digested with Liberase TL and DNase I as previously stated. Digested lymph nodes were then processed into single-cell suspensions. 2500 beads from the Trucount Absolute Counting Tubes (BD Bsiocienes) were added to each sample to later determined the total cell-count number. Then, cells were stained for viability (Live/Dead Blue), and Fc receptor blocking together with CD8β for 20 min at 4 °C. Then, the samples were stained for the rest of surface markers, including the H-2K$^b$ OVA$_{257-264}$ (SIINFEKL) tetramer for 30 min at 4 °C. Samples were kept in ice until acquisition by flow cytometry no more than 4 h later. Cell numbers were calculated by normalizing the values by multiplying by the ratio of added/read Trucount beads.

## Statistical analysis

Unless stated otherwise, all statistical analyses were performed with GraphPad Prism 8.0 (GraphPad Software, 758 San Diego, CA, USA) and represented as mean ± standard error of the mean (SEM). The n represents the number of biological replicates. Unless otherwise specified, statistical analyses were performed using unpaired two-tailed Student's *t*-test, one-way analysis of variance (ANOVA) with a Tukey post-test, or two-way ANOVA with Sidak's multiple comparisons tests. Boxplot graphs represent the median and expand from 10- to 90-percentil, with error bars from min to max. Differences with *p*-values < 0.05 were considered significant: ns, non-significant; *$p < 0.05$; **$p < 0.01$; ***$p < 0.001$; ****$p < 0.0001$. Cells drawings in figures were obtained from Servier Medical Art, provided by Servier, licensed under a Creative Commons Attribution 3.0 unported license.

## Reporting summary

Further information on research design is available in the Nature Portfolio Reporting Summary linked to this article.

## Data availability

The mass spectrometry proteomics data have been deposited to the ProteomeXchange Consortium via the PRIDE partner repository with the dataset identifier PXD039035 and is publicly available as of the date of publication (https://proteomecentral.proteomexchange.org/cgi/GetDataset?ID=PXD039035). All other data are available in the article and its Supplementary files or from the corresponding author upon request. This paper does not report original code. Source data are provided with this paper.

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

## Acknowledgements

This study was supported by the Spanish Ministry of Science and Innovation (grants PID2020-120412RB-I00, PDC2021-121797-I00, PGC2018-097019-BI00, PID2021-122348NB-I00, PLEC2022-009235, PLEC2022-009298, PID2021-125415OB-I00, and PID2019-105761RB-I00); Comunidad de Madrid (INTEGRAMUNE, P2022/BMD7209 and IMMUNO-VAR, P2022/BMD-7333); Ramón Areces Foundation "Ciencias de la Vida y la Salud" (XIX Concurso-2018); "la Caixa" Banking Foundation (grants HR17-00016, HR17-00247, and HR22-00253); ProteoRed from Instituto de Salud Carlos III (PT17/0019/0003- ISCIII-SGEFI / ERDF); CIBER Cardiovascular (CB16/11/00272, CB16/11/00277); Agencia Estatal de Investigación (AEI); Fondo de Investigación Sanitaria del Instituto de Salud Carlos III; co-funding by Fondo Europeo de Desarrollo Regional (FEDER); and European Research Council Starting Grant SYNVIVO 853179. D.C.-F. is supported by an INPhINIT Retaining Fellowship from "la Caixa"

Foundation (LCF/BQ/DR19/11740010). S.I. is supported by a RYC-2016-19463 fellowship. E.H. is supported by an FPI fellowship (PRE2019-087509). We thank Miguel Vicente-Manzanares for critically reading the manuscript. The CNIC is supported by the Instituto de Salud Carlos III (ISCIII), the Ministerio de Ciencia e Innovación (MCIN) and the Pro CNIC Foundation, and is a Severo Ochoa Center of Excellence (grant CEX2020-001041-S funded by MICIN/AEI/10.13039/501100011033). The QIAGEN IPA software was used to create Figs. 3a and 5a.

## Author contributions

D.C.-F. designed conceptualized, performed experiments, analysis, data interpretation, made figures and wrote the manuscript. S.I. performed experiments, design, conceptualization, and data interpretation. I.J. performed proteomics experiments and analysis. M.R.-H. and E.H.-G. experiments and data collection. E.D. performed and analyzed ex vivo LIPSTIC experiments. N.M.-C. helped performing Seahorse and confocal microscopy experiments. E.N.-V. produced IAV. E.V. produced Listeria stocks and provided β2m-/- mice. J.V. supervised proteomics studies and performed statistical analysis. G.P. performed experiments, design, data analysis and conceptualization of LIPSTIC experiments. F.S.-M. raised funding, supervised, and revised all the work.

## Competing interests

G.P. is co-inventor of the US patent US10053683 about LIPSTIC technology. The remaining authors declare no competing interests.
