## [Peer Review File · Nature Communications]

Immune synapse drives lipid peroxidation and MHC-I upregulation in licensed dendritic cells for efficient priming of CD8+ T cellsREVIEWER COMMENTS

Reviewer #1 (Remarks to the Author):

The manuscript describes changes in DC following the establishment of immunological synapses with CD4 T cells. Some of the changes improve the antigen cross-presentation ability of the DC, resulting in enhanced CD8 T cell priming. They describe these results as a component of the process of “licensing”.

Overall, the results in the manuscript are convincing and well presented. Several approaches are innovative and clever, enabling validation in vivo of conclusions of experiments carried out in vitro. I believe my comments can be satisfactorily addressed in a revised version within a reasonable timeframe.

1. At some point the authors should clarify that GM-CSF-induced BMDC do not represent the “conventional DC” found in vivo. They are better described as “monocyte derived DC (or just cells)”. This is not a major problem because the authors validate their conclusions using this cell culture with cDC in vivo.
2. The authors hypothesize reasonable mechanistic explanations for the increased cross-presenting activity of licensed DC eg lipid peroxidation. They do not formally demonstrate these mechanisms but I do not think this limitation is severe enough to preclude publication.
3. Experiment in Fig. 1G-K requires clarification. If the DC were incubated with OVA protein, wouldn't they be performing MHC II presentation even if they are not incubated with the peptide? Is the different outcome caused by the incubation with peptide a matter of increased presentation/recognition. If yes, would they conclude the phenomenon they describe is dependent on the avidity of CD4 T cell recognition? This is a matter that should be discussed.
4. Has the role of lipid droplets reported in ref #32 been validated in independent studies?

Signed: as a matter of principle, I believe peer review should always be anonymous.

Reviewer #2 (Remarks to the Author):

The manuscript studies the impact of antigen-specific interaction between DC and CD4 T cells on DC protein expression and the DC ability to stimulate CD8 T cell immune responses. While studying DC maturation is not novel, the bioinformatic analysis of proteome shifts on maturation is a strong tool to study the impact of CD4 interaction. The finding of changes in lipid peroxidation is a result of this analysis, which are clearly described and mechanistic experiments demonstrate that it is impactful to T cell activation.

The most interesting technique in the manuscript uses the LIPSTIC approach to label DC that had interacted with CD4 T cells as per Pasqual et al 2018. The key data here demonstrates that the effect of CD4 interactions on MHCI and lipid peroxidation is limited to those that interacted with CD4 T cells. This is an elegant model and its use adds to the literature.

The major issues with the manuscript are the infection models, using intranasal DC followed 24hrs later by intranasal flu, or intravenous DC followed 24hrs later by intravenous *Listeria monocytogenes*. For the flu model there are no controls to demonstrate any impact of DC versus no DC transfer, to understand whether the psDC or the nsDC represent no impact or a change from baseline. This seems necessary since the authors need to demonstrate that DC administered intranasally can function normally to reach the same locations as T cells. If intranasal DC cross-present infectious antigen but cannot cross the nasal/lung mucosa, this model has limited value.

The route of DC migration in the IV *Listeria* model is again biologically questionable, since the matured DC must presumably transmigrate through to the spleen for interaction with *Listeria*-specific CD8 T cells. In this IV model DC transfer has a very obvious impact since the IV injection of psDC or nsDC has a dramatic impact on *Listeria*-induced toxicity even in Rag^{-/-} mice (Figure S4). These data suggest that the model is poorly suited to test their hypothesis since there is such a strong T cell independent action. In particular, Figure 4G showing that when the DC lack the ability to present to CD8 T cells they are dramatically effective at protecting from *Listeria* challenge, if not different between psDC and nsDC, shows that this model is not appropriately demonstrating a T cell mechanism. Protection is better in Figure 4G when DC cannot present to T cells than Figure 4B, despite the data in Figure 4E.

Summarizing the above two points, at present the data in the infectious models do not provide sufficient support for the central hypothesis and limit the value of the work.

The final experiments, in Figure 6 are very clever. However, they do not strongly support their arguments. Specifically, depletion of the cells that have interacted with CD4 T cells using the biotin-LPETG leaves a strong response to antigen. This suggests that the licensed DC are not particularly important. Despite this, the text (line 333) states: “Hence, we found that psDCs are responsible for the generation of antigen-specific CD8+ T cells responses during immunization in vivo.” The data shows a response without these cells, so this statement is false.

In relation to the above, the changes in T cell responses to intranasal flu are fairly marginal. These data show that while psDC transfer makes slightly better flu antigen-specific responses, they are very similar to the response with nsDC transfer. Again, it is not clear that this model shows a particular importance for CD4-matured DC when applied in vivo.

Aside from these issues, the manuscript does support the main claims – that CD4 interactions improve DC maturation and this results in altered lipid peroxidation.

There are number of minor issues that should be addressed:

Figure 4 D, F, H (spelling weight) – same in supplementals.

In various places, for example line 303, the authors state that the psDC generate a ‘broader’ effector response. This is not demonstrated. It might be semantic, but the data shows a ‘larger’ response. Breadth of response would need a different approach to demonstrate the inclusion of multiple different populations, whether different clones or different phenotypes, not more of one type.

Line 272. “Increased percentage of CD8”. These data show CD8 as a percentage of CD3. We do not know whether this represents more CD8 in total, or for example a loss of CD4 T cells.

These data would be better represented as percentage of live cells, or even better as absolute numbers as is shown elsewhere (for example Figure 6).

Related to the above. CD8 percentage increase, CD4 percentage do not change (Line 286 Figure S5E). What decreases? Something must since we are showing percent of CD3. Is there a change in the CD3+ CD4-CD8- population? Are these gd T cells or NKT? Are these relevant to the response.

Figure S5H appears to show a large population of CD44-CD62L- CD8 T cells. The plots in S5A are much more as expected. What is the identity of the CD44-CD62L- CD8 T cells?

Figure S5I Tem in CD8. Left axis needs correcting to "freq in CD8"?

In Figure 5, what conclusions are made from the MFI of the tetramer+ cells? Is this an assumption of affinity? It is not clear that this measure is valuable without other analysis such as peptide titrations or additional phenotyping.

Reviewer #3 (Remarks to the Author):

In this report Calzada-Fraile et al investigate the cellular and molecular mechanisms underpinning the ability of DCs that have interacted with CD4+ T cells ("postsynaptic" DCs, psDCs) to generate CD8+ T cell responses. Using an unbiased, proteomics-based approach they show that, following antigen-specific interaction with cognate CD4+ T cells, psDCs undergo a proteomic remodelling that shows a bias towards pathways implicated in antigen processing and presentation via MHCI. This bias is reflected by an enhanced ability of psDCs to prime CD8+ T cells through antigen cross-presentation. Interestingly, the proteomic analysis also highlighted an upregulation in psDCs of the "accumulation of lipids" as well as "dysregulation of redox homeostasis" categories, which they functionally link by showing an enhanced lipid peroxidation in these cells that they found to be required for the licensing function of psDCs for cross-presentation. The upregulation in MHCI expression and lipid peroxidation in psDCs was confirmed both ex vivo and in vivo during immunization, using a robust mouse model that allows proximity-labelling with biotin of recipient cells (DCs) that

have closely interacted with donor cells (CD4+ T cells). Consistent with this function, they show that licenced psDCs protect mice for bacterial infection in a CD8+ T cell-dependent manner following adoptive transfer and enhance the generation of flu-specific CD8+ effector and memory cells following intranasal inoculation. Finally, they show that depletion of interacting DC:T cells during in vivo immunization results in the abrogation of antigen-specific CD8+ T cell responses.

This report provides fundamental new insights into the mechanisms that underpin the licencing of DCs to induce productive CD8+ T cell responses to infection through a flow of information from CD4+ T cells to DCs occurring at their interface. The report elegantly explores DC licencing, starting from an unbiased high throughput approach that revealed not only a mechanism that could be expected -upregulation of the antigen processing and presentation pathway, but a completely new mechanism involving lipid peroxidation. These findings are then translated to the in vivo context of antigen-specific responses and infection, using powerful mouse models to investigate antigen specificity and to discriminate in vivo DCs that have interacted with CD4+ T cells from the other DCs. The article is very clearly written and the data are very rigorous. I only have two questions that could be addressed experimentally and highlight some points that should be clarified.

Point 1. The authors describe a bystander effect of psDCs not only on CD8+ T cells specific for the antigen recognized by the CD4+ T cells responsible for their licencing, but also on other CD8+ T cells (lines 326-329 and 404-406). Can the authors discuss about how this is achieved? In the context of an immune response there will be both DCs that have interacted with CD4+ T cells in an antigen-specific manner and other DCs. Do they propose that the cross-presentation function on CD8+ T cells is carried out uniquely by psDC? And would the mechanism involve the "superior" antigen presentation properties of psDCs? Do they think that this bystander function requires only their physical interaction with CD8+ T cells or also soluble factors released by psDCs?

Point 2. Although licencing has been shown to be mediated by CD40 induction, it would be useful if the role of CD40 would be experimentally tested in the settings used in this report.

Point 3. The authors refer to the formation of immune synapses between CD4+ T cells and DCs. Although this is reasonably expected, the actual formation of an immune synapse has not been investigated. A different wording would be preferable.

Point 4. Lines 283 and 303. The authors refer to a "broader" CD8+ T cell response. This can be misleading as the term usually applies to the array of antigens against which a response is generated. A different wording would be preferable.

Point 5. Lines 143-144. The authors report a decrease glycoytic activity of psDCs compared to nsDCs, however in figure S2D the profiles appear very similar. Can the authors comment? Also, a statistical should be provided (also for panel C).

Point 6. Line 200. Please add the reference after "as previously reported".

Reviewer #1 (Remarks to the Author):

The manuscript describes changes in DC following the establishment of immunological synapses with CD4 T cells. Some of the changes improve the antigen cross-presentation ability of the DC, resulting in enhanced CD8 T cell priming. They describe these results as a component of the process of “licensing”.

Overall, the results in the manuscript are convincing and well presented. Several approaches are innovative and clever, enabling validation *in vivo* of conclusions of experiments carried out *in vitro*. I believe my comments can be satisfactorily addressed in a revised version within a reasonable timeframe.

1. At some point the authors should clarify that GM-CSF-induced BMDC do not represent the “conventional DC” found *in vivo*. They are better described as “monocyte derived DC (or just cells)”. This is not a major problem because the authors validate their conclusions using this cell culture with cDC *in vivo*.

We thank reviewer #1 for the suggestion. We have included this clarification in the manuscript in page 8, second paragraph: “BMDCs are a useful tool to study licensing of antigen presenting cells². They share with DCs isolated from tissues the ability to present antigens to T cells and to respond to microbial stimuli through maturation, but they are considered as surrogates of *in vivo*-occurring conventional DCs (cDCs)^{37,38}”.

2. The authors hypothesize reasonable mechanistic explanations for the increased cross-presenting activity of licensed DC eg lipid peroxidation. They do not formally demonstrate these mechanisms but I do not think this limitation is severe enough to preclude publication.

We thank reviewer #1 for this observation. According to this suggestion, we have softened the mechanistic claims from “mediated” to “associated with” in page 8, first paragraph: “Therefore, a dysregulation of the redox homeostasis on psDCs resulted in increased levels of lipid peroxidation, which associated with the increased ability of psDCs to cross-present soluble protein to CD8+ T cells”.

3. Experiment in Fig. 1G-K requires clarification. If the DC were incubated with OVA protein, wouldn't they be performing MHC II presentation even if they are not incubated with the peptide? Is the different outcome caused by the incubation with peptide a matter of increased presentation/recognition. If yes, would they conclude the phenomenon they describe is dependent on the avidity of CD4 T cell recognition? This is a matter that should be discussed.

We thank reviewer #1 for this question. Indeed, as all DCs are incubated with OVA protein they would perform MHC II presentation even if they are not incubated with the peptide. This is why, in this case, the DCs which are incubated with OVA protein and without OT-II peptide are not cocultured with CD4+ T cells. In this way we control that no antigen presentation is taking place on the nsDC sample although the MHC II molecules of these DCs may be loaded with OTII peptides derived from OVA protein processing. Therefore, we can conclude that we are comparing psDC samples to DC samples in which no antigen presentation took place (nsDCs). Following this suggestion, we have clarified the experimental conditions in the manuscript under materials and methods section (page 31): “Then, OT-II CD4+ T cells or media were added to the psDC and nsDC sample respectively at a 1:2 (DC:CD4) ratio and cocultured overnight”.

4. Has the role of lipid droplets reported in ref #32 been validated in independent studies?

We thank reviewer #1 for the question. Indeed, there are other reports that claim that the induction of cross-presentation on DCs by saponin and carbomer-based adjuvants is dependent on lipid droplet formation. As these studies support ref #32, we added these references in the manuscript (ref #33 and #34 in page 6).

Signed: as a matter of principle, I believe peer review should always be anonymous.

Reviewer #2 (Remarks to the Author):

The manuscript studies the impact of antigen-specific interaction between DC and CD4 T cells on DC protein expression and the DC ability to stimulate CD8 T cell immune responses. While studying DC maturation is not novel, the bioinformatic analysis of proteome shifts on maturation is a strong tool to study the impact of CD4 interaction. The finding of changes in lipid peroxidation is a result of this analysis, which are clearly described and mechanistic experiments demonstrate that it is impactful to T cell activation.

The most interesting technique in the manuscript uses the LIPSTIC approach to label DC that had interacted with CD4 T cells as per Pasqual et al 2018. The key data here demonstrates that the effect of CD4 interactions on MHC I and lipid peroxidation is limited to those that interacted with CD4 T cells. This is an elegant model and its use adds to the literature.

The major issues with the manuscript are the infection models, using intranasal DC followed 24hrs later by intranasal flu, or intravenous DC followed 24hrs later by intravenous *Listeria monocytogenes*. For the flu model there are no controls to demonstrate any impact of DC versus no DC transfer, to understand whether the psDC or the nsDC represent no impact or a change from baseline. This seems necessary since the authors need to demonstrate that DC administered intranasally can function normally to reach the same locations as T cells. If intranasal DC cross-present infectious antigen but cannot cross the nasal/lung mucosa, this model has limited value.

We thank reviewer #2 for raising these points. Indeed, we had carried out experiments in which the transfer of DCs was compared to the transfer of PBS as a control condition. We had performed these experiments while setting up the infection model with X31, an H3N2 influenza virus that causes mild to moderate illness in mice, instead of the mouse-adapted H1N1 PR8 virus. Once we determined that the effect was significantly different from the baseline (PBS) (see added graphs to Figure S5B-D below), we performed the experiments shown in Figure 5 and S5 maximizing the number of animals per condition treated with psDCs and nsDCs. Therefore, these results suggest that the transfer of psDCs represents an induction of influenza-specific CD8+ T cells compared to the baseline (PBS), which does not differ from nsDCs. We have included these results in Figure S5B-D and indicated it in the manuscript in page 13.

Moreover, although the transfer of DCs represents an induction of specific CD8+ T cells compared to the baseline, we addressed, following reviewer's #2 indication, the biodistribution of injected DCs. For this, we injected CFSE-labelled CD45.1+ psDCs or nsDCs into CD45.2+ mice i.n. or i.v. and analyzed the presence of transferred DCs in the lung, mediastinal lymph node, and spleen 24h after the DC transfer.

As a result of this biodistribution profile, we found the i.v. or i.n. transferred DCs in the lung, in accordance with what was previously described (Firdessa-Fite and Creusot, 2020). In agreement with an increased homing ability of psDCs (Alcaraz-Serna et al., 2021), increased numbers of psDCs vs nsDCs were recovered. We have no evidence that transferred DCs reach mediastinal lymph nodes, the expected secondary lymphoid organ to which intranasally-administered DCs may have migrated to. Similarly, DCs transferred i.v. could not be detected in the spleen. However, their transfer has a clear effect on inducing CD8+ T cells compared to the baseline, which is the main concern raised by reviewer #2 to prove that the effect is driven by the DCs. We have included this data in the manuscript in page 12, figure S4F, and methodology in page 35.

Although we have no evidence that the transferred DCs are reaching the secondary lymphoid organs assessed, they could be presenting antigen by: (i) reaching untested lymphoid organs; (ii) being involved not in the first priming of CD8+ T cells in secondary lymphoid organs, but rather in the activation and expansion of CD8+ T cells that egress from these organs or are circulating systemically and/or reach the tissue; or (iii) priming naïve T cells directly in the lung parenchyma. Naïve T cells can be found in lung parenchyma (Cose et al., 2006) and have been shown to migrate from lung to lymph nodes (Caucheteux et al., 2013). Moreover, primary activation of CD8 T cells in non-secondary lymphoid organs has been previously shown (Bertolino et al., 2001). This is discussed in page 20.

To show that transferred DCs can be directly responsible for the generation of influenza-specific CD8+ T cells, we transferred psDCs into Batf3^{-/-} mice, which lack cross-presenting cDC1s and display impaired CD8+ T cell responses during influenza infection (Waithman et al., 2013), and evaluated the IAV-specific CD8+ T cell response in peripheral blood 9 days after infection. As shown in the figure below (included as Figure S5K), psDC transfer increased the proportion of Tetramer+ CD8+CD44+ T cells, indicating that the transfer of licensed psDC were able to

promote IAV-specific CD8⁺ T cells in the absence of endogenous cross-presenting DCs. We have included these new data in the manuscript in page 13-14, Figure S5K, and discussed it in page 20.

Alcaraz-Serna, A., Bustos-Morán, E., Fernández-Delgado, I., Calzada-Fraile, D., Torralba, D., Marina-Zárate, E., Lorenzo-Vivas, E., Vázquez, E., Albuquerque, J.B. de, Ruef, N., Gómez, M.J., Sánchez-Cabo, F., Dopazo, A., Stein, J.V., Ramiro, A., Sánchez-Madrid, F., 2021. Immune synapse instructs epigenomic and transcriptomic functional reprogramming in dendritic cells. *Sci. Adv.* 7, eabb9965. <https://doi.org/10.1126/sciadv.abb9965>

Bertolino, P., Bowen, D.G., McCaughan, G.W., Fazekas de St. Groth, B., 2001. Antigen-Specific Primary Activation of CD8⁺ T Cells Within the Liver1. *J. Immunol.* 166, 5430–5438. <https://doi.org/10.4049/jimmunol.166.9.5430>

Caucheteux, S.M., Torabi-Parizi, P., Paul, W.E., 2013. Analysis of naïve lung CD4 T cells provides evidence of functional lung to lymph node migration. *Proc. Natl. Acad. Sci.* 110, 1821–1826. <https://doi.org/10.1073/pnas.1221306110>

Cose, S., Brammer, C., Khanna, K.M., Masopust, D., Lefrançois, L., 2006. Evidence that a significant number of naive T cells enter non-lymphoid organs as part of a normal migratory pathway. *Eur. J. Immunol.* 36, 1423–1433. <https://doi.org/10.1002/eji.200535539>

Firdessa-Fite, R., Creusot, R.J., 2020. Nanoparticles versus Dendritic Cells as Vehicles to Deliver mRNA Encoding Multiple Epitopes for Immunotherapy. *Mol. Ther. Methods Clin. Dev.* 16, 50–62. <https://doi.org/10.1016/j.omtm.2019.10.015>

Waithman, J., Zanker, D., Xiao, K., Oveissi, S., Wylie, B., Ng, R., Tögel, L., Chen, W., 2013. Resident CD8⁺ and Migratory CD103⁺ Dendritic Cells Control CD8 T Cell Immunity during Acute Influenza Infection. *PLOS ONE* 8, e66136. <https://doi.org/10.1371/journal.pone.0066136>

The route of DC migration in the IV *Listeria* model is again biologically questionable, since the matured DC must presumably transmigrate through to the spleen for interaction with *Listeria*-specific CD8 T cells. In this IV model DC transfer has a very obvious impact since the IV injection of psDC or nsDC has a dramatic impact on *Listeria*-induced toxicity even in Rag^{-/-} mice (Figure S4). These data suggest that the model is poorly suited to test their hypothesis since there is such a strong T cell independent action. In particular, Figure 4G showing that when the DC lack the ability to present to CD8 T cells they are dramatically effective at protecting from *Listeria* challenge, if not different between psDC and nsDC, shows that this model is not appropriately demonstrating a T cell mechanism. Protection is better in Figure 4G when DC cannot present to T cells than Figure 4B, despite the data in Figure 4E.

We thank reviewer #2 for this observation. Regarding the route of administration and the transmigration through the spleen, we believe that the fact that the effect is directly driven by the transferred DCs is properly addressed in the previous point. Moreover, we performed an intravenous delivery model, causing systemic *Listeria* infection. Therefore, the transferred DCs may have better access to pathogen antigens and

interaction with CD8⁺ T cells, which could also be occurring in other infection-relevant organs (Bertolino et al., 2001).

Regarding the protection observed in Figure 4, we agree with the referee and mention in the manuscript that there is a protection effect provided by the transfer of LPS-activated DCs alone. However, in our opinion, this does not disqualify this model to test our hypothesis, as a clear further protection provided by psDCs vs nsDCs can be observed amid the protection provided by the DC transfer itself. As both psDCs and nsDCs have a similar T-cell independent effect shown in Figure S4A, the differences between them observed in Figure 4B rely on an adaptive dependent effect.

This T cell-independent effect is also observed in Figure 4G. As reviewer states, protection in this case seems to be greater than in Figure 4B, despite transferred DCs not having the ability to present to CD8⁺ T cells. However, it is clear that the increased protection effect of wild type psDCs is dependent on their ability to activate CD8⁺ T cells as this difference with nsDCs is lost in this case. The fact that protection is greater than in Figure 4B, which was as well surprising to us, may be due to: (i) the transfer of cells without MHC class I molecules may be activating NK cells that aid in the protection observed in Figure 4G, (ii) the reduced ability to generate an adaptive response may be translating into a reduced immunopathology that leads to reduced mortality during the infection, or (iii) the variability of the model, in which we observe variable death rates in different experiments despite using the same bacterial dose. Although of great interest because this scenario further increases the protection effect, we believe that the reason behind the increased protection in Figure 4G it is out of the scope of this work and does not invalidate the observation that the difference between psDCs and nsDCs is lost when transferring $\beta 2m^{-/-}$ cells. We appreciate this comment raised by the reviewer and have discussed it in the manuscript in page 19-20.

Summarizing the above two points, at present the data in the infectious models do not provide sufficient support for the central hypothesis and limit the value of the work.

The final experiments, in Figure 6 are very clever. However, they do not strongly support their arguments. Specifically, depletion of the cells that have interacted with CD4 T cells using the biotin-LPETG leaves a strong response to antigen. This suggests that the licensed DC are not particularly important. Despite this, the text (line 333) states: "Hence, we found that psDCs are responsible for the generation of antigen-specific CD8⁺ T cells responses during immunization in vivo." The data shows a response without these cells, so this statement is false.

We thank reviewer #2 for the observation. Indeed, abrogation of antigen-specific CD8⁺ T cells is not complete, but it shows a ca. 50% reduction, something that is even more prominent (reduction to almost 1/3) when looking at CD44⁺CD62L⁻ CD8⁺ T cells. This reduction may not be complete because, as the reviewer suggests, psDC may not be fully responsible of antigen-specific CD8⁺ T cell generation. Hence, following the reviewer's observation, we have softened our claim and included this observation in page 15 and 16: "Strikingly, when interacting cells had been biotinylated, SA-Saporin administration resulted in a significant 50% reduction in the number of SIINFEKL-specific CD8⁺ T cells (Figure 6B)" [...] "Hence, we found that psDCs are partly responsible for the generation of antigen-specific CD8⁺ T cells responses during immunization in vivo".

In relation to the above, the changes in T cell responses to intranasal flu are fairly marginal. These data show that while psDC transfer makes slightly better flu antigen-specific responses, they are very similar to the response with nsDC transfer. Again, it is not clear that this model shows a particular importance for CD4-matured DC when applied in vivo.

Aside from these issues, the manuscript does support the main claims – that CD4 interactions improve DC maturation and this results in altered lipid peroxidation.

We hope that reviewer #2 finds that our new data further supports and increases the value of our work, as we believe it is now clearer that transferred DCs are directly responsible for the increased generation of pathogen-specific CD8+ T cell responses. As explained above, this is supported in the evidence showing that that (1) the transfer of DCs has a biological effect on the generation of CD8+ T cell responses compared to the treatment with PBS; (2) although transferred DCs are mostly found in the lung when injected both intravenously and intranasally, these DCs can expand T cells previously activated in secondary lymphoid organs or directly activate naïve T cells; and more importantly (3) the transfer of psDCs into Batf3 KO mice, which do not efficiently generate antigen-specific CD8+ T cell responses during infection, partly rescues the generation of these responses, indicating that, in this model, transferred DCs are inducing influenza-specific CD8+ T cell responses.

There are number of minor issues that should be addressed:

Figure 4 D, F, H (spelling weight) – same in supplementals.

We thank the reviewer for the observation. We have corrected these misspellings.

In various places, for example line 303, the authors state that the psDC generate a 'broader' effector response. This is not demonstrated. It might be semantic, but the data shows a 'larger' response. Breadth of response would need a different approach to demonstrate the inclusion of multiple different populations, whether different clones or different phenotypes, not more of one type.

Following reviewer #2's suggestion, we have changed the wording of psDC inducing a broader CD8+ T cell response "promoting" or "enhancing" CD8+ T cell responses in page 12 and 13.

Line 272. "Increased percentage of CD8". These data show CD8 as a percentage of CD3. We do not know whether this represents more CD8 in total, or for example a loss of CD4 T cells. These data would be better represented as percentage of live cells, or even better as absolute numbers as is shown elsewhere (for example Figure 6).

We thank the reviewer for the suggestion. We do not have the total count number for this experiment. However, we do have the data of CD8+ T cells as freq of live cells and these data and the differences observed are equivalent to the freq. of CD3+ T cells, both for CD8 and CD4 T cells (see figure below). Following reviewer's suggestion, we have indicated in the manuscript that the same is observed for frequencies of both subsets in live cells (page 13).

Related to the above. CD8 percentage increase, CD4 percentage do not change (Line 286 Figure S5E). What decreases? Something must since we are showing percent of CD3. Is there a change in the CD3⁺ CD4⁻CD8⁻ population? Are these gd T cells or NKT? Are these relevant to the response.

We thank reviewer #2 for the question. As reported in the previous point, there is no change in the CD4⁺ population within live cells or CD3⁺ cells. However, we neither observe a change in the CD3⁺CD4⁻CD8⁻ population (see figure below). As we observe no changes, this population, which can be composed of NKT cells or $\gamma\delta$ T cells as the reviewer suggests, does not seem to be relevant to this model.

Figure S5H appears to show a large population of CD44⁻CD62L⁻ CD8⁺ T cells. The plots in S5A are much more as expected. What is the identity of the CD44⁻CD62L⁻ CD8⁺ T cells?

We thank reviewer #2 for the question. Indeed, this population has been previously observed, but is still poorly understood. Previous reports define this CD44⁻CD62L⁻ subset as “pre-effector-like T cells” that appear in secondary lymphoid organs. This came from the observation that stimulating *in vitro* naive CD8⁺ T cells first generated CD44⁻CD62L⁻ cells, followed by effector/memory T cells (Nakajima et al., 2021). Moreover, in another report, the expansion of APCs *in vivo* by injection of Flt3L induced the expansion of CD44⁻CD62L⁻ CD8⁺ T cells specifically in spleen and lymph nodes (Wolf et al., 2022). This could be similar to what is happening in our system, as we are also incrementing the number of APCs by injection of the DCs. Hence, these

reports are in agreement with the fact that we observe this population in the spleen, a secondary lymphoid organ (Figure S5L), but not in peripheral blood (Figure S5A).

Nakajima, Y., Chamoto, K., Oura, T., Honjo, T., 2021. Critical role of the CD44^{low}CD62L^{low} CD8⁺ T cell subset in restoring antitumor immunity in aged mice. *Proc. Natl. Acad. Sci.* 118, e2103730118. <https://doi.org/10.1073/pnas.2103730118>

Wolf, G., Gerber, A.N., Fasana, Z.G., Rosenberg, K., Singh, N.J., 2022. Acute effects of FLT3L treatment on T cells in intact mice. *Sci. Rep.* 12, 19487. <https://doi.org/10.1038/s41598-022-24126-4>

Figure S5I Tem in CD8. Left axis needs correcting to “freq in CD8”?
We thank the reviewer for the observation. We have corrected this.

In Figure 5, what conclusions are made from the MFI of the tetramer+ cells? Is this an assumption of affinity? It is not clear that this measure is valuable without other analysis such as peptide titrations or additional phenotyping.

We thank reviewer #2 for the question. In Figure 5D, G and S5G we show the Tetramer MFI within the total indicated population, not within the Tetramer+ population. Hence, the increase in tetramer fluorescence is an indirect indication of more tetramer+ cells in the population complementary to the tetramer+ frequency graphs shown in Figure 5C, F and S5F.

Reviewer #3 (Remarks to the Author):

In this report Calzada-Fraile et al investigate the cellular and molecular mechanisms underpinning the ability of DCs that have interacted with CD4⁺ T cells ("postsynaptic" DCs, psDCs) to generate CD8⁺ T cell responses. Using an unbiased, proteomics-based approach they show that, following antigen-specific interaction with cognate CD4⁺ T cells, psDCs undergo a proteomic remodelling that shows a bias towards pathways implicated in antigen processing and presentation via MHC I. This bias is reflected by an enhanced ability of psDCs to prime CD8⁺ T cells through antigen cross-presentation. Interestingly, the proteomic analysis also highlighted an upregulation in psDCs of the "accumulation of lipids" as well as "dysregulation of redox homeostasis" categories, which they functionally link by showing an enhanced lipid peroxidation in these cells that they found to be required for the licensing function of psDCs for cross-presentation. The upregulation in MHC I expression and lipid peroxidation in psDCs was confirmed both ex vivo and in vivo during immunization, using a robust mouse model that allows proximity-labelling with biotin of recipient cells (DCs) that have closely interacted with donor cells (CD4⁺ T cells). Consistent with this function, they show that licensed psDCs protect mice for bacterial infection in a CD8⁺ T cell-dependent manner following adoptive transfer and enhance the generation of flu-specific CD8⁺ effector and memory cells following intranasal inoculation. Finally, they show that depletion of interacting DC:T cells during in vivo immunization results in the abrogation of antigen-specific CD8⁺ T cell responses.

This report provides fundamental new insights into the mechanisms that underpin the licensing of DCs to induce productive CD8⁺ T cell responses to infection through a flow of information from CD4⁺ T cells to DCs occurring at their interface. The report elegantly explores DC licensing, starting from an unbiased high throughput approach that revealed not only a mechanism that could be expected -upregulation of the antigen processing and presentation pathway, but a completely new mechanism involving lipid peroxidation. These findings are then translated to the in vivo context of antigen-specific responses and infection, using powerful mouse models to investigate antigen specificity and to discriminate in vivo DCs that have interacted with CD4⁺ T cells from the other DCs. The article is very clearly written and the data are very rigorous. I only have two questions that could be addressed experimentally and highlight some points that should be clarified.

Point 1. The authors describe a bystander effect of psDCs not only on CD8⁺ T cells specific for the antigen recognized by the CD4⁺ T cells responsible for their licensing, but also on other CD8⁺ T cells (lines 326-329 and 404-406). Can the authors discuss about how this is achieved? In the context of an immune response there will be both DCs that have interacted with CD4⁺ T cells in an antigen-specific manner and other DCs. Do they propose that the cross-presentation function on CD8⁺ T cells is carried out uniquely by psDC? And would the mechanism involve the "superior" antigen presentation properties of psDCs? Do they think that this bystander function requires only their physical interaction with CD8⁺ T cells or also soluble factors released by psDCs?

We thank reviewer #3 for these questions. As the depletion of interacting T:DC pairs decreases not only SIINFEKL-specific CD8⁺ T cells, but also the general CD8⁺CD44⁺ T cell population, we postulated that it could be affecting other responding CD8⁺ T cells. Since SIINFEKL is not the only peptide from OVA protein that is presented to

CD8⁺ T cells (Karandikar et al., 2019), the effect of reduction in total CD8⁺CD44⁺ T cells is the sum of the all OVA-specific CD8⁺ T cells (responsive to SIINFEKL and other OVA peptides). Moreover, as the reviewer suggests, we cannot rule out the possibility that there is activation of bystander CD8⁺ T cells by e.g. cytokines. Indeed, IL27 has been recently reported to be produced by interacting DCs in the LIPSTIC system and this promotes T cell activation. Therefore, we have reflected these possibilities in the manuscript in page 15-16: “This indicates that depletion of interacting DC:T cell pairs abrogates the generation, not only of SIINFEKL-specific CD8⁺ T cells, but may also affect other OVA-specific CD8⁺ T cells. Additionally, we cannot rule out the activation of bystander CD8⁺ T cells by other means such as cytokine secretion⁴⁴.”

Finally, as the reviewer states, there may be other DCs that have not interacted with CD4⁺ T cells and that may be able to cross-present. However, here we are only depleting DCs that have interacted with CD4⁺ T cells as these are the only ones that are biotinylated in this system. Hence, the effect that we observe is due to the ability of interacting DCs cross-presenting to CD8⁺ T cells. As this effect is not a complete abrogation since there may be other DCs that can cross-present, we have softened our claim: “Hence, we found that psDCs are partly responsible for the generation of antigen-specific CD8⁺ T cells responses during immunization in vivo.”

Karandikar, S.H., Sidney, J., Sette, A., Selby, M.J., Korman, A.J., Srivastava, P.K., 2019. Identification of epitopes in ovalbumin that provide insights for cancer neoepitopes. *JCI Insight* 4. <https://doi.org/10.1172/jci.insight.127882>

Point 2. Although licencing has been shown to be mediated by CD40 induction, it would be useful if the role of CD40 would be experimentally tested in the settings used in this report.

We thank reviewer #3 for this interesting question. Following reviewer #3's question, we have blocked the CD40-CD40L interaction during antigen-cognate interactions and assessed the effect on the changes for MHC-I and lipid peroxidation on primary psDC vs. nsDCs (see figure below). We have performed this experiment using wild type primary splenic DCs and not the LIPSTIC system as in Figure 3A because blocking the CD40-CD40L interaction was previously reported to abrogate the biotin labelling in the LIPSTIC system (Pasqual et al.). The differences of MHC-I levels between psDCs and nsDCs were not abrogated although the amount of MHC-I molecules significantly decreased in both psDCs and nsDCs when CD40-CD40L interactions were blocked during synaptic interactions. Remarkably, lipid peroxidation significantly decreased in psDCs when blocking CD40 induction by CD40L, displaying lower lipid peroxidation than nsDCs. Surprisingly, the effect of blocking CD40L antibody in nsDCs during co-culture with CD4 T cells is clearly different from the one observed in psDCs, since these nsDCs generated in the presence of anti-CD40L showed the most increased levels of lipid peroxidation. We have included and discussed these new results (see below) in the manuscript in page 10 and 17-18 and Figure S3J-L.

Point 3. The authors refer to the formation of immune synapses between CD4+ T cells and DCs. Although this is reasonably expected, the actual formation of an immune synapse has not been investigated. A different wording would be preferable.

We thank reviewer #3 for the observation. As the reviewer states, an immune synapse (IS) in this condition is expected, as previously reported by T cell activation in the psDC fraction but not in the nsDC fraction (Alcaraz-Serna et al. 2021. Science Advances). Hence, we may assume IS formation in this system. At any rate, we have experimentally demonstrated the formation of IS by confocal microscopy and, as it can be observed, IS appears in psDC but not nsDC samples as observed by the polarization of actin cytoskeleton in T cells, the morphology of the contact, and the polarization of microtubules and the centrosome towards the IS site in psDC and not in nsDC samples. Having experimentally proven the formation of an immune synapse in psDCs and not in nsDCs, we considered to keep the wording of immune synapse. We have included this result in the manuscript in page 4, Figure S1A, and methodology in page 27-28. We also include below a 3D reconstruction of the results.

Point 4. Lines 283 and 303. The authors refer to a "broader" CD8+ T cell response. This can be misleading as the term usually applies to the array of antigens against which a response is generated. A different wording would be preferable.

Following reviewer #3's suggestion, we have changed the wording of psDC inducing a broader CD8+ T cell response "promoting" or "enhancing" CD8+ T cell responses since, as the reviewer says, we do not know whether the CD8+ T cell response is broader in the sense of responsive to a bigger diversity of antigens.

Point 5. Lines 143-144. The authors report a decrease glycoytic activity of psDCs compared to nsDCs, however in figure S2D the profiles appear very similar. Can the authors comment? Also, a statistical should be provided (also for panel C).

We thank the reviewer for the observation. Indeed, we have written it wrongly. As stated by the reviewer, Seahorse ECAR (Figure S2D) is equal in both cell types and this indicates no differences in glycolysis as predicted by proteomics (Figure S2A). However, the OCR indicates a decreased activity of the mitochondria in psDC. We have amended the text accordingly in page 7 and added a statistic to the OCR data as there are no significant differences in ECAR: "Seahorse XF analysis revealed a decreased activity of the mitochondria, but no differences in glycolytic activity of psDCs (Figure S2C-D), as the proteomic dataset indicated (Figure S2A)."

Point 6. Line 200. Please add the reference after "as previously reported". Following this suggestion, we have included the reference following the sentence.

REVIEWERS' COMMENTS

Reviewer #1 (Remarks to the Author):

The authors have thoroughly addressed my concerns and the other reviewers'. I have no further comments.

Reviewer #2 (Remarks to the Author):

The authors have been responsive to review and have clarified a number of important points. This reviewer still has concerns about the infectious models, but it is sufficient that these are raised and discussed in the manuscript.

One sticking point is the use of MFI in Figure 5. With the clarification provided by the authors that MFI is being used to identify the size of the antigen specific (tetramer+) population, this is not the correct approach. MFI should not be used on unseparated bimodal data, and is only appropriate to described shifts in homogenous populations. Percent positive is the correct measure here.

Otherwise, this reviewer has no further issues that need addressing.

Reviewer #3 (Remarks to the Author):

The authors have addressed satisfactorily all the issues raised in my previous review.

REVIEWERS' COMMENTS

Reviewer #1 (Remarks to the Author):

The authors have thoroughly addressed my concerns and the other reviewers'. I have no further comments.

We thank reviewer #1 for the comments.

Reviewer #2 (Remarks to the Author):

The authors have been responsive to review and have clarified a number of important points. This reviewer still has concerns about the infectious models, but it is sufficient that these are raised and discussed in the manuscript.

One sticking point is the use of MFI in Figure 5. With the clarification provided by the authors that MFI is being used to identify the size of the antigen specific (tetramer+) population, this is not the correct approach. MFI should not be used on unseparated bimodal data, and is only appropriate to described shifts in homogenous populations. Percent positive is the correct measure here.

Otherwise, this reviewer has no further issues that need addressing.

We thank reviewer #2 for the comments. According to the suggestion, we have removed the tetramer MFI data in Figure 5 (now Figure 6) and Figure S5 and left only the tetramer+ proportions to indicate the size of antigen-specific populations.

Reviewer #3 (Remarks to the Author):

The authors have addressed satisfactorily all the issues raised in my previous review.

We thank reviewer #3 for the comments.